# META-PREDICTION MODEL FOR DISTILLATION-AWARE NAS ON UNSEEN DATASETS

**Hayeon Lee**[*]  **Sohyun An**[*]  **Minseon Kim**,  **Sung Ju Hwang**
KAIST, South Korea
{hayeon926, sohyunan, minseonkim, sjhwang82}@kaist.ac.kr

## ABSTRACT

Distillation-aware Neural Architecture Search (DaNAS) aims to search for an optimal student architecture that obtains the best performance and/or efficiency when distilling the knowledge from a given teacher model. Previous DaNAS methods have mostly tackled the search for the neural architecture for fixed datasets and the teacher, which are not generalized well on a new task consisting of an unseen dataset and an unseen teacher, thus need to perform a costly search for any new combination of the datasets and the teachers. For standard NAS tasks without KD, meta-learning-based computationally efficient NAS methods have been proposed, which learn the generalized search process over multiple tasks (datasets) and transfer the knowledge obtained over those tasks to a new task. However, since they assume learning from scratch without KD from a teacher, they might not be ideal for DaNAS scenarios. To eliminate the excessive computational cost of DaNAS methods and the sub-optimality of rapid NAS methods, we propose a distillation-aware meta accuracy prediction model, **DaSS** (**D**istillation-**a**ware **S**tudent **S**earch), which can predict a given architecture's final performances on a dataset when performing KD with a given teacher, without having actually to train it on the target task. The experimental results demonstrate that our proposed meta-prediction model successfully generalizes to multiple unseen datasets for DaNAS tasks, largely outperforming existing meta-NAS methods and rapid NAS baselines. Code is available at https://github.com/CownowAn/DaSS.

## 1 INTRODUCTION

Distillation-aware Neural Architecture Search (DaNAS) aims to search for an optimal student architecture that obtains the best performance and efficiency on a given dataset when distilling the knowledge from the given teacher to it (Liu et al., 2020; Gu & Tresp, 2020; Kim et al., 2022). For the DaNAS task, we need to design a framework that considers the effect of Knowledge Distillation (KD), yet, conventional NAS frameworks may be sub-optimal as they do not consider KD components at all by searching for an architecture according to its evaluations trained from scratch. As explained in Liu et al. (2020), the sub-optimality of conventional NAS methods on DaNAS tasks results from: 1) For the same target dataset, an optimal student architecture for distilling the knowledge from the teacher and an optimal student architecture for learning from scratch with only ground-truth labels may be different. 2) Even for the same dataset, the optimal student architecture may depend on the specific teacher. To tackle such challenges, existing DaNAS methods guide the search process using the KD loss (Liu et al., 2020) or propose a proxy to evaluate distillation performance (Kim et al., 2022).

However, such existing DaNAS methods do not generalize to multiple tasks, require training for any combination of dataset and teachers, and may result in excessive computational cost (e.g., 5 days with 200 TPUv2, for each task (Liu et al., 2020)). This hinders their applications to real-world scenarios since optimal student architectures may vary depending on the type of datasets, teacher, and resource budgets. Therefore, we need a rapid and lightweight DaNAS method that can be generalized across different settings.

For standard NAS tasks without KD, there has been some progress in the development of rapid NAS methods that are computationally efficient, such as 1) meta-learning-based transferable NAS

---

[*]These authors contributed equally to this work.

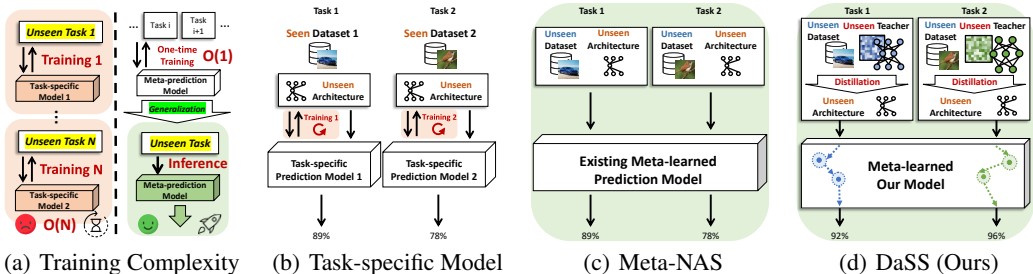

| (a) Training Complexity | (b) Task-specific Model | (c) Meta-NAS | (d) DaSS (Ours) |

Figure 1: **Concept.** To search for a student architecture optimized for a distillation task, the prediction model should estimate the final accuracy of architecture differently depending on a dataset, a teacher network, a student architecture, and a distillation process. While existing meta-prediction models only support set-conditioned prediction, the proposed meta-prediction model, DaSS performs the distillation-task-conditioned prediction.

methods (Lee et al., 2021a;b; Jeong et al., 2021) and 2) zero-cost proxies (Mellor et al., 2021; Abdelfattah et al., 2021). The former meta-NAS methods learn the generalized search process over multiple tasks, allowing it to adapt to a novel unseen task by transferring the knowledge obtained over the meta-learning phase to the new task without training the NAS framework from scratch. To this end, meta-NAS methods generally utilize a task-adaptive prediction model, which rapidly adapts to novel datasets and devices. They outperform baseline NAS methods on multiple benchmark datasets (Lee et al., 2021a) and real-world datasets (Jeong et al., 2021), as well as with various devices (Lee et al., 2021b), significantly reducing the architecture search time to less than a few GPU seconds on an unseen setting. The latter, zero-cost NAS methods have proposed several proxies that can be obtained from the first mini-batch without fully training the architecture on the target dataset.

However, despite the success of such rapid NAS methods on standard NAS tasks, they may be sub-optimal for DaNAS scenarios since they assume training from scratch, not KD from a teacher, which may significantly impact the actual accuracy of the architecture retrieved from the search. Therefore, to overcome the high search cost of DaNAS methods and the sub-optimality of rapid NAS methods, we propose a rapid distillation-aware meta-prediction model, **DaSS** (**D**istillation-**a**ware **S**tudent **S**earch), for DaNAS tasks (Fig. 1). Following the previous works on meta-NAS, we leverage meta-learning to learn a prediction model that can rapidly adapt to an unseen target task. However, our approach has two main differences.: 1) Distillation-aware design of the meta-prediction model and 2) a meta-learning scheme that utilizes already trained teachers, both of which are optimized for the DaNAS task. First, we propose a distillation-aware task encoding function that considers the output from the student whose parameters are remapped from the teacher to estimate the teacher's impact on the actual performance of the distilled student network. Second, we use the accuracy of the teacher to guide the gradient-based adaptation of the meta-prediction model. This allows a more accurate and rapid estimation of the architecture's performance on a target task (dataset) with a specific teacher.

We meta-learn the proposed distillation-aware prediction model on the subsets of TinyImageNet and neural architectures from the ResNet search space. Then we validate its prediction performance on heterogeneous unseen datasets such as CUB, Stanford Cars, DTD, Quickdraw, CropDisease, EuroSAT, ISIC, ChestX, and ImageNet-1K. The experimental results show that our meta-learned prediction model adapts to novel target tasks to estimate the actual performance of an architecture distilled by an unseen teacher within 3.45 (wall clock sec) on average without direct training on the target tasks. Further, the DaNAS framework with the proposed distillation-aware meta-prediction model outperforms existing meta-NAS and zero-cost proxies on the same set of datasets.

To summarize, our contributions in this work are as follows:

- We propose a novel meta-prediction model, DaSS, that generalizes across datasets, architectures, and teachers, which can accurately predict the performance of an architecture when distilling the knowledge of the given teacher.

- We propose a novel distillation-aware task encoding based on the functional embeddings of a specific teacher and parameter-remapped student architecture candidates.

- We enable a rapid gradient-based one-shot adaptation of the meta-prediction model on a target task by guiding it with a teacher-accuracy pair.

- We experimentally validate that our meta-prediction model successfully estimates the actual accuracies of the student architectures for a given teacher on multiple unseen datasets, significantly outperforming existing methods in the correlation between prediction and GT accuracies and in the end-to-end DaNAS performances.

## 2 RELATED WORK

**Distillation-aware NAS**    Knowledge Distillation (KD) is an efficient model compression technique under the resource budgets by transferring the knowledge from the large teacher to the smaller student network. Since distilling performance depends on architectures as verified in Liu et al. (2020), it is important to search for an optimal student architecture given the dataset and teacher. Recently, while Neural Architecture Search (NAS) suggests automating designing the optimal architecture (Zoph & Le, 2017; Baker et al., 2017; Zhong et al., 2018; Real et al., 2017; Liu et al., 2018; Elsken et al., 2019; Real et al., 2019; Liu et al., 2019) that aims to search for an optimal architecture that has high accuracy when trained with only the ground-truth label, it might be suboptimal for searching for an architecture optimized for distilling given the teacher. Liu et al. (2020) verified the assumption with the RL agent, which KD-trains sampled architecture for a few epochs to obtain a KD-guided accuracy proxy. It takes 5 days with 200 TPUv2 on a single task, so it cannot be applied to real-world KD scenarios. Kim et al. (2022) suggested the trust region with a Bayesian optimization search algorithm and a new KD-guided score. However, it cannot be generalized to multiple tasks, which makes rapid search impossible. To obtain a pruned student architecture, (Gu & Tresp, 2020) specify scaling factors for all channels of a teacher and optimize that with KD-aware loss function applying a structured pruning method. As the optimization should be performed repeatedly for each task to get the optimal student architecture, the total time is still too large.

**Meta-learning and Meta-prediction Model**    Meta-learning (learning to learn) (Thrun & Pratt, 1998) allows a model to quickly adapt to an unseen task, by learning to generalize over the distribution of tasks and transferring the meta-knowledge to the unseen task. One of the main meta-learning approaches is a gradient-based meta-learning (Finn et al., 2017; Li et al., 2017), which learns an initialization parameter shared across multiple tasks to be optimal for any tasks within a few gradient-based updates from the initial parameter. Recently, meta-learning-based NAS (Meta-NAS) frameworks (Lee et al., 2021a; Jeong et al., 2021; Lee et al., 2021b) have been proposed, which are rapid and transferable to large-scale unseen tasks. MetaD2A (Lee et al., 2021a) and TANS (Jeong et al., 2021) designed the accuracy prediction models and Lee et al. (2021b) proposed the latency prediction model. MetaD2A and TANS proposed the set encoder for task-adaptive accuracy prediction models. However, none of them do consider the teacher and distillation process, thereby an architecture obtained by them might be suboptimal when distilling knowledge from the teacher to the architecture. Due to the high cost and time to collect architecture-accuracy pairs as samples for adaptation on unseen tasks, they do not leverage the advantage of the few-shot adaptation on a target task. On the other hand, we utilize a teacher-accuracy pair for adaptation, allowing the predictive model to do gradient-based adaptation to new tasks. The additional required time is the time for a forward pass to obtain the accuracy of the teacher.

## 3 BACKGROUND

This section describes existing accuracy prediction models designed for NAS and their limitations in using the models rapidly across multiple tasks of searching for optimal student architectures to distill the knowledge from teacher networks. In Table 1, we briefly summarize differences between task-specific accuracy prediction models (Task-spec.), existing meta-accuracy prediction models (MetaD2A (Lee et al., 2021a), TANS (Jeong et al., 2021)), and the proposed meta-accuracy prediction model for DaSS (Ours). As the task-specific model needs to be trained on each task, the total training time increases as the

Table 1: Comparison between accuracy prediction models for NAS and DaNAS.

|  | Task-spec. | MetaD2A | TANS | **DaSS (Ours)** |
|---|---|---|---|---|
| Time Compl. | $O(N)$ | $O(1)$ | $O(1)$ | $O(1)$ |
| Meta-learning | X | O | O | **O** |
| Unseen Arch. | O | O | O | **O** |
| Unseen Data. | X | O | O | **O** |
| Unseen Teach. | X | X | X | **O** |
| Distil-aware. | X | X | X | **O** |
| Fast Adapt. | X | X | X | **O** |

number of tasks increases ($O(N)$). Since meta-prediction models including ours can be generalized to multiple unseen tasks (datasets) after one-time meta-training, the training time is constant even if the number of tasks increases ($O(1)$).

However, while ours is generalized for unseen teacher networks (Unseen Teach.) and estimate distillation-aware accuracy (Distil-aware.), existing meta-prediction models have difficulty doing them. Furthermore, we empirically demonstrate the efficacy of the gradient-based one-shot fast adaptation with teacher-accuracy pair to improve the prediction performance of the meta-prediction model on an unseen task in NAS (Fast Adapt.).

**Task-specific Accuracy Prediction Model** During the search process, training each architecture candidate on a task until convergence to obtain its final performance, which is a single-scalar value, is wasteful. Thus, many NAS methods (Luo et al., 2018; Wang et al., 2020; Cai et al., 2020) leverage a task-specific accuracy prediction model as a proxy to reduce the excessive search cost spent on training the models. Let us assume a task specification $\tau = \{\mathbf{D}^\tau, \mathbf{S}, \mathbf{Y}^\tau\}$ where $\mathbf{D}^\tau \subset \mathcal{D}$ is a dataset, $\mathbf{S} \subset \mathcal{S}$ is a set of neural architectures, and $\mathbf{Y}^\tau \subset \mathcal{Y}$ is a set of accuracies obtained by training each architecture $s \in \mathbf{S}$ on the dataset $\mathcal{D}^\tau$. The task-specific accuracy prediction model $f^\tau(s; \phi) : \mathcal{S} \to \mathbb{R}$ parameterized by $\phi$ is trained for each task $\tau$ that estimates the accuracy $y^\tau \in \mathbf{Y}^\tau$ of a neural architecture $s \in \mathbf{S}$ for a given dataset $\mathbf{D}^\tau$ by minimizing empirical loss, $\mathcal{L}$ (e.g. mean squared error) defined on the predicted values $f^\tau(\mathbf{s}; \phi)$ and actual measurements $y^\tau$ as follows:

$$\min_{\phi} \mathcal{L}\big(f^\tau(\mathbf{S}; \phi), \mathbf{Y}^\tau\big) \tag{1}$$

 However, the task-specific accuracy prediction model is highly limited in its practical application to a real-world setting. Since it will not generalize to a new task (dataset), we need to learn a task-specific accuracy prediction model on the new target dataset. However, this is highly time-consuming since we need to collect many architecture-accuracy pairs (at least thousands of samples) by training sampled architectures on the new dataset.

**Meta-accuracy Prediction Model** To address such inefficiency in task-specific NAS and enable transferrable NAS on multiple tasks, Lee et al. (2021a) and Jeong et al. (2021) proposed generalizable accuracy prediction models that can rapidly adapt to unseen datasets **within a few GPU seconds** after a single round of meta-learning. Specifically, such rapid search is made possible by meta-learning the prediction model conditioned on the given dataset: $f(s, \mathbf{D}^\tau; \phi) : \mathcal{S} \times \mathcal{D} \to \mathbb{R}$. This enables it to support **multiple** datasets with a **single** prediction model since the predicted accuracy $y \leftarrow (s, \mathbf{D}^\tau)$ is dependent on both the dataset type $\mathbf{D}^\tau$ and the architecture $s$. Then, they train the prediction model to generalize over the task distribution $p(\tau)$ by sampling task $\tau$ per iteration via amortized meta-learning as follows:

$$\min_{\phi} \sum_{\tau \sim p(\tau)} \mathcal{L}\big(f(\mathbf{S}, \mathbf{D}^\tau; \phi), \mathbf{Y}^\tau\big) \tag{2}$$

After this one-time meta-learning, the set-conditioned accuracy prediction model $f(\cdot; \phi^*)$ with meta-learned parameters $\phi^*$ can be transferred to an unseen task $\hat{\tau}$ to search for an architecture optimized for $\hat{\tau}$. Note that the meta-learned prediction model works well on an unseen dataset $\hat{\mathbf{D}}^{\hat{\tau}}$ which has no overlap with datasets $\{\mathbf{D}^\tau\}$ used in the meta-training phase. This reduces the meta-prediction model's complexity from $O(N)$ to $O(1)$ on $N$ tasks.

**Limitations** While the meta-accuracy prediction models are promising as they enable rapid search on multiple tasks, directly using the existing meta-accuracy prediction model for distillation-aware NAS problems is suboptimal. This is because they do not consider the teacher and may be inaccurate for DaNAS scenarios due to the following reasons:

1. **Distillation-independent accuracy prediction.** DaNAS is different from the existing NAS problems in that a task $\tau$ in the distillation-aware NAS domain includes a teacher $\mathbf{t}(\cdot; \tilde{\boldsymbol{\theta}})^\tau$ which is parameterized by $\tilde{\boldsymbol{\theta}}$ and is trained on a dataset $\mathbf{D}^\tau$. Thus, for the same dataset, the optimal student architectures can be different if the teachers are different. However, the existing meta-prediction model does not encode the teacher and thus will search for the same student architecture for different teachers, which might be sub-optimal. Thus, we need a teacher-conditioned accuracy prediction model for DaNAS.

2. **No gradient-guided task adaptation.** Existing Meta-NAS frameworks aim to learn a generalized accuracy prediction for multiple tasks with amortized meta-learning conditioned on the task embeddings, not leveraging the power of gradient-based adaptation steps (Finn

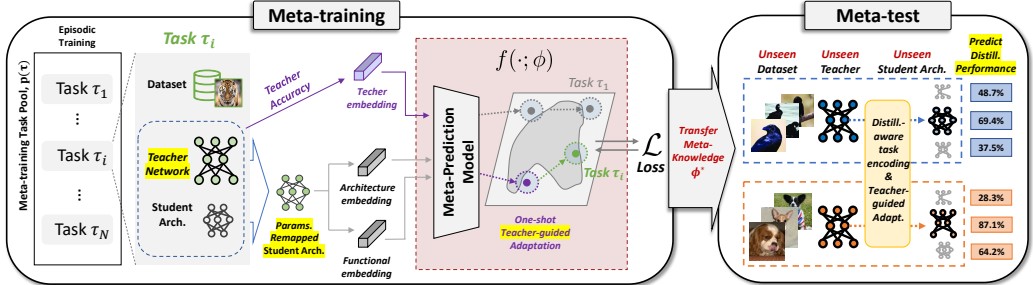

Figure 2: **Overview of DaSS.** To search for an optimal student architecture that can distill the knowledge of given teacher $\mathbf{t}^\tau(\cdot; \mathbf{D}^\tau, \tilde{\boldsymbol{\theta}}^\tau)$ well, the sampled task should be encoded in a distillation-aware way. We first remap the trained parameters of the teacher to the randomly sampled student. Then, by taking a fixed input to the teacher and the remapped student, we can encode the teacher and student functionally in a distillation-aware way. Along with the functional embedding, by considering the architecture configuration of the student, we encode sampled tasks in a way appropriate for the DaNAS. By transferring meta-knowledge $\phi^*$ learned from meta training phase, our meta-prediction model $f(\cdot; \phi^*)$ can adapt to an unseen task $\hat{\tau}$ quickly through the teacher-guided adaptation with almost no cost.

et al., 2017; Li et al., 2017). This was mainly done to expedite the meta-training step and reduce the adaptation time on a new task since gradient-based adaptation requires obtaining few-shot architecture-accuracy samples of the target dataset, which takes at least a few GPU min to hours and renders rapid NAS impossible.

To address the above issues, we aim to design a teacher-conditioned accuracy prediction model and the accuracy of teacher-guided meta-learning for rapid DaNAS tasks. In our work, we reveal that teacher-conditioned accuracy prediction is more accurate than set-conditioned accuracy prediction when estimating actual performances of student architectures in DaNAS scenarios. In addition, we show that it is possible to perform gradient-guided task adaptation using the accuracy of the teacher. Please refer to Fig. 2, which illustrates our distillation-aware performance prediction model, DaSS.

## 4 METHOD

**Distillation-aware Accuracy Prediction Model**    For each task $\tau = \{\mathbf{S}, \mathbf{Y}^\tau, \mathbf{t}^\tau(\cdot; \mathbf{D}^\tau, \tilde{\boldsymbol{\theta}}^\tau)\}$ from a distillation-aware NAS domain, we propose a distillation-aware accuracy meta-prediction model $f(\cdot; \phi)$ parameterized by $\phi$ as follows:

$$f(s, \mathbf{t}^\tau(\cdot; \mathbf{D}^\tau, \tilde{\boldsymbol{\theta}}^\tau); \phi) : \mathcal{S} \times \mathcal{T} \to \mathbb{R} \tag{3}$$

that can estimate an accuracy of a student architecture $s$ differently depending the student architecture $s$ and the teacher $\mathbf{t}^\tau(\cdot; \mathbf{D}^\tau, \tilde{\boldsymbol{\theta}}^\tau)$ which is trained on the target dataset, $\mathbf{D}^\tau$. We encode a task $\tau$ by utilizing the teacher which is trained on the target dataset and the student architecture in a distillation-aware way. To this end, we encode each sampled student architecture candidate $s(\cdot; \boldsymbol{\theta})$ in a teacher-adaptive manner by leveraging a parameter remapping $\tilde{\boldsymbol{\theta}}_s^\tau = g(\tilde{\boldsymbol{\theta}}^\tau)$ with the remapping function $g$ from the parameters $\tilde{\boldsymbol{\theta}}^\tau$ of the teacher trained on the dataset $\mathbf{D}^\tau$ to the parameters $\tilde{\boldsymbol{\theta}}_s^\tau$ of each student architecture $s(\cdot; \tilde{\boldsymbol{\theta}}_s^\tau)$, which can be done at almost no cost. More detailed descriptions about the parameter remapping function, $g$ is in Appendix B. Next, we embed each remapped student candidate $s(\cdot; \tilde{\boldsymbol{\theta}}_s^\tau)$ and the teacher $\mathbf{t}^\tau(\cdot; \tilde{\boldsymbol{\theta}}^\tau)$ by feeding in a fixed Gaussian random noise $\mathbf{z}$ (Jeong et al., 2021) as input as follows:

$$s(\mathbf{z}; \tilde{\boldsymbol{\theta}}_s^\tau) = \boldsymbol{h}_{z,s}^\tau, \quad \mathbf{t}^\tau(\mathbf{z}; \tilde{\boldsymbol{\theta}}^\tau) = \boldsymbol{h}_{z,t}^\tau \tag{4}$$

In addition, as described in Appendix A, our ResNet search space is factorized hierarchical search space, and all architecture candidates in this search space are composed of $4$ stages. Student architecture candidates in this search space can have various depths and channel widths at each stage. As suggested in Cai et al. (2020), the architecture configuration, such as depth and channel widths at each stage of $s$, can be represented as a one-hot encoding vector. Concatenating one-hot encoding vectors for all stages, we can obtain the embedding for the whole network architecture, $\boldsymbol{h}_a$. The accuracy prediction model takes three embeddings $\boldsymbol{h}_a$, $\boldsymbol{h}_{z,s}^\tau$ and $\boldsymbol{h}_{z,t}^\tau$ as inputs and outputs the estimated accuracy, with additional nonlinear projection layers. For the detailed descriptions, please refer to Appendix B.

**Adaptation Guided with Teacher-Accuracy Pair** To obtain meta-learned $f(\cdot; \boldsymbol{\phi}^*)$, we meta-learn the proposed accuracy prediction model over task samples from the task distribution $p(\tau)$ as follows:

$$\min_{\boldsymbol{\phi}} \sum_{\tau \sim p(\tau)} \mathcal{L}\big(f\big(\mathbf{S}, \mathbf{t}^\tau(\cdot; \mathbf{D}^\tau, \tilde{\boldsymbol{\theta}}^\tau); \boldsymbol{\phi}\big), \mathbf{Y}^\tau\big) \tag{5}$$

where $\mathcal{L}$ is the Mean Square Error loss. Differently from the previous meta-accuracy prediction models (Lee et al., 2021a; Jeong et al., 2021) based on the amortized meta-learning, we further perform a guided adaptation on the target task with the teacher $\boldsymbol{t}^\tau$ and its accuracy $\tilde{y}^\tau$ evaluated on the target dataset, by conducting the inner gradient updates with them as follows:

$$\boldsymbol{\phi}_{(i+1)}^\tau = \boldsymbol{\phi}_{(i)}^\tau - \alpha \nabla_{\boldsymbol{\phi}_{(i)}} \mathcal{L}\big(f\big(\mathbf{t}^\tau, \mathbf{t}^\tau(\cdot; \mathbf{D}^\tau, \tilde{\boldsymbol{\theta}}^\tau); \boldsymbol{\phi}_{(i)}\big), \tilde{y}^\tau\big) \quad \text{for } i = 1, \dots, I \tag{6}$$

where $i$ denotes the $i$-th inner gradient step, $I$ is the total number of inner gradient steps, and $\alpha$ is the multi-dimensional inner learning rate vector (Li et al., 2017). This meta-learning scheme enables the accuracy prediction model to start from the knowledge learned from the meta-training tasks and rapidly adapt to the new task with guidance from the teacher-accuracy pair. Note that while there are gradient-based meta-learning methods (Finn et al., 2017; Li et al., 2017; Lee et al., 2021b), our work is the first to train the meta-accuracy prediction model for distillation-aware NAS that can generalize to unseen datasets via gradient-based guided adaptation. Furthermore, by introducing the simple yet effective approach of using the teacher and its accuracy pair as few-shot training instances for task adaptation, we significantly reduce the time-consuming collection of the student architecture-accuracy pairs for each task, which allows efficient adaptation to each task. The final meta-learning objective reflecting Eq. (5) and Eq. (6) is given as follows:

$$\min_{\boldsymbol{\phi}} \sum_{\tau \sim p(\tau)} \mathcal{L}\big(f\big(\mathbf{S}, \mathbf{t}^\tau(\cdot; \mathbf{D}^\tau, \tilde{\boldsymbol{\theta}}^\tau); \boldsymbol{\phi}_{(I+1)}^\tau\big), \mathbf{Y}^\tau\big) \tag{7}$$

**Adapting the Meta-prediction Model on Unseen Tasks (Meta-test)** In Fig. 2, we utilize the meta-learned prediction model $f(\cdot; \boldsymbol{\phi}^*)$ on a novel distillation task $\hat{\tau}$ that was unseen during meta-training. Given a task $\hat{\tau} = \{\mathbf{D}^{\hat{\tau}}, \mathbf{S}, \mathbf{Y}^{\hat{\tau}}, \mathbf{t}^{\hat{\tau}}(\cdot; \tilde{\boldsymbol{\theta}}^{\hat{\tau}})\}$, we first compute the accuracy $\tilde{y}^{\hat{\tau}} = \mathbf{t}^{\hat{\tau}}(\mathbf{D}^{\hat{\tau}}; \tilde{\boldsymbol{\theta}}^{\hat{\tau}})$ of the teacher through the forward process on the validation. Then, we update the meta-learned parameters $\boldsymbol{\phi}_{(1)}^{\hat{\tau}} = \boldsymbol{\phi}^*$ by adapting it to the unseen task by taking inner gradient steps, using Eq. (6) to obtain $\boldsymbol{\phi}_{(I+1)}^{\hat{\tau}}$ with $\{\mathbf{t}^{\hat{\tau}}(\cdot; \mathbf{D}^{\hat{\tau}}, \tilde{\boldsymbol{\theta}}^{\hat{\tau}}), \tilde{y}^{\hat{\tau}}\}$. Then, for each student architecture candidate $s$, we remap parameters and compute $\boldsymbol{h}_{z,t}^{\hat{\tau}}$ and $\boldsymbol{h}_{z,s}^{\hat{\tau}}$ with Eq. (4) and obtain $\boldsymbol{h}_a^{\hat{\tau}}$. Finally, we concatenate $\boldsymbol{h}_{z,s}^{\hat{\tau}}, \boldsymbol{h}_a^{\hat{\tau}}$ and $\boldsymbol{h}_{z,t}^{\hat{\tau}}$ and compute the estimated accuracy of $s$ on the task $\hat{\tau}$ with $f(\boldsymbol{h}_{z,t}^{\hat{\tau}}, \boldsymbol{h}_{z,s}^{\hat{\tau}}, \boldsymbol{h}_a^{\hat{\tau}}; \boldsymbol{\phi}_{(I+1)}^{\hat{\tau}})$.

## 5 EXPERIMENT

We first validate and analyze our meta-prediction model, DaSS, in terms of 1) a novel distillation-aware task encoding and 2) a gradient-based one-shot adaptation strategy with the teacher by predicting the distillation performance of unseen student architectures on an unseen dataset with an unseen teacher in Section 5.1. Second, we report the time efficiency of DaSS compared with few-shot meta-prediction models in Section 5.2. Next, we demonstrate the performance of DaSS on end-to-end DaNAS tasks compared against existing rapid NAS methods in Section 5.3. In addition to this, we compared the performance of ours and the existing Meta-NAS methods through end-to-end DaNAS tasks on cross-domain datasets in Section 5.4. Lastly, we validated DaSS on ImageNet-1K which is one of the representative large-scale datasets with $256 \times 256$ pixels in Section 5.5. A detailed description of our search space throughout the experiments is in Appendix A, and we use the largest architecture in our search space as teacher architecture for all experiments.

**Meta-Training Database** Following Ryu et al. (2020), we use TinyImageNet (Le & Yang, 2015) as the source dataset for meta-training. We generate a heterogeneous set of tasks by class-wisely splitting the dataset into 10 splits. We use subsets consisting of 8 and 2 splits for meta-training and meta-validation, respectively. Please refer to Appendix D for the more detailed explanation.

**Generalization to Unseen Datasets** We transfer the meta-knowledge of the prediction model trained over the meta-training phase to meta-test tasks. Each meta-test task consists of an unseen

Table 2: **Efficacy of DaSS for DaNAS task.** We report Spearman's rank correlation coefficient between ranking by actual distillation performance and ranking by predicted distillation performance of 50 unseen student architectures on an unseen dataset, with an unseen teacher. The reported values are the average over 3 random runs. Our meta-prediction model with teacher-guided meta-learning strategy outperforms all rapid meta-prediction models.

| Method | Model Components | | | | Spearman's Rank Correlation Coefficient | | | | |
| | Arch. | Set Enc. | Distill-aware. | Guide | CUB | Cars | DTD | Draw | Mean |
|---|---|---|---|---|---|---|---|---|---|
| MAML (Finn et al., 2017) | ✓ | | | ✓ | 0.63 | 0.61 | 0.38 | 0.42 | 0.51 |
| Meta-SGD (Li et al., 2017) | ✓ | | | ✓ | 0.18 | 0.49 | 0.19 | 0.63 | 0.37 |
| MetaD2A (Lee et al., 2021a) | ✓ | ✓ | | | 0.63 | 0.51 | 0.40 | 0.26 | 0.45 |
| TANS (Jeong et al., 2021) | ✓ | ✓ | | | 0.43 | 0.61 | -0.15 | -0.17 | 0.18 |
| DaSS (Ours, Non-Guide) | ✓ | | ✓ | | 0.69 | 0.77 | 0.50 | 0.62 | 0.64 |
| **DaSS (Ours, Guide)** | ✓ | | ✓ | ✓ | **0.69** | **0.88** | **0.57** | **0.75** | **0.72** |

dataset, an unseen teacher trained on the unseen dataset, and multiple unseen student architecture candidates. We covered 9 datasets, including fine-grained datasets (CUB (Wah et al., 2011), Stanford Cars (Krause et al., 2013) (Cars), CropDisease (Guo et al., 2020) - agriculture images), out-of-distribution datasets (DTD (Cimpoi et al., 2014) - texture images, Quick Draw (Jonas et al.) (Draw) - black-and-white drawing images, EuroSAT (Guo et al., 2020) - satellite images, ISIC (Guo et al., 2020) - medical images about skin lesions, ChestX (Guo et al., 2020) - X-ray images), and large-scale dataset - ImageNet-1K (Russakovsky et al., 2015).

## 5.1 EFFICACY OF THE PROPOSED META-PREDICTION MODEL ON UNSEEN DATASETS

We validate whether our novel distillation-aware task encoding and meta-learning strategy, which is a gradient-based one-shot adaptation with the teacher, help improve the performance of the accuracy prediction model. We use fixed 50 student architectures and distill knowledge from the teacher on each unseen dataset to compute Spearman's rank correlation (the higher, the better) between the predicted and actual performances and report the efficacy of each method. In Table 2, **Arch.** means taking architecture configuration, **Set Enc.** means taking dataset embedding, **Distill-aware.** means taking the functional embeddings of student architecture candidate and teacher, and **Guide** means the adaptation strategy guided by teacher-accuracy pair in a meta-learning scheme.

DaSS (Ours, Guide) outperforms all rapid meta-prediction models (Finn et al., 2017; Li et al., 2017; Lee et al., 2021a; Jeong et al., 2021) on 4 unseen datasets. MAML and Meta-SGD, which learn accuracy prediction models by taking only the architecture configuration (Arch.) of a student candidate as an input, show poor performance, as they predict the distillation performance without consideration of a target dataset and teacher. Set-conditioned (Set Enc.) prediction models (MetaD2A and TANS) also achieve significantly lower performance than ours since they search architectures in a teacher-agnostic manner, which greatly impacts the performance of the student trained with distillation. On the other hand, using the proposed distillation-aware embeddings of a student and a teacher (Distill-aware.), DaSS (Ours, Guide) significantly outperforms all baselines on all 4 unseen tasks. Furthermore, we observe that adaptation to the target task by guiding our meta-prediction model with the accuracy of the teacher on the target dataset largely improves the prediction performance over the non-guided search, DaSS (Ours, Non-Guide). In sum, the results show the clear advantage of both the design of our distillation-aware task encoding and the teacher-guided meta-learning scheme.

## 5.2 EFFICIENCY OF TEACHER GUIDED ACCURACY PREDICTION

In general, guided search, such as few-shot adaptation on the target task, requires the collection of few-shot architecture-accuracy pairs. To obtain even a single pair of architecture-accuracy, we should train the architecture until it fully converges, which prevents rapid search at meta-test time. However, the DaNAS environment is different from the conventional NAS in that we already have a trained model, the **teacher**. There is

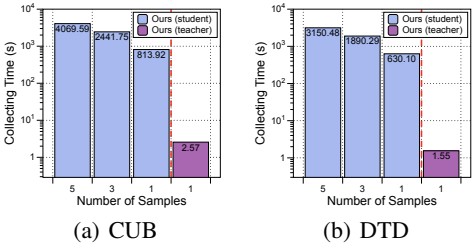

(a) CUB          (b) DTD

Figure 3: **Time and Sample Efficiency.**

no need to wait to get the accuracy-architecture pair(s) for adaptation since we can get the accuracy of the teacher with a single forward pass. In Fig. 3, we report the time efficiency of DaSS, compared against that of the baselines, when adapting to unseen tasks with student architecture-accuracy pair(s). The **Collecting time** refers to the time required to obtain architecture-performance pairs (**Number of samples**) in the guided meta-learning strategy. Our one-shot adaptation guided search with the teacher network which is already trained under the DaNAS task is much more efficient than utilizing a few-shot adaptation with a few student architecture-accuracy pair(s). Our model only needs $2.57(s)$ and $1.55(s)$ on **CUB** and **DTD**, respectively, which significantly reduces the adaptation time.

## 5.3 Comparison with Rapid NAS Methods

Table 3: The reported results are **SRC** between ranking by actual distillation performance and ranking by predicted distillation performance of unseen student architectures on an unseen dataset, with an unseen teacher. **Search Time** is the time for predicting the performance of each unseen student architecture by given method. Concretely, the search time represents the total search time divided by the number of architecture candidates (e.g., search time per architecture).

| Method | Spearman's Rank Correlation Coefficient (SRC) | | | | | Search Time / Arch. (GPU sec) | | | | |
| | CUB | Cars | DTD | Draw | Mean | CUB | Cars | DTD | Draw | Mean |
|---|---|---|---|---|---|---|---|---|---|---|
| **Zero-cost proxies** | | | | | | | | | | |
| Grad Norm (Abdelfattah et al., 2021) | -0.07 | 0.09 | 0.19 | -0.60 | -0.09 | 0.98 | 1.00 | 1.11 | 2.11 | 1.30 |
| Snip (Abdelfattah et al., 2021) | 0.11 | 0.27 | 0.24 | -0.81 | -0.04 | 1.05 | 1.03 | 0.97 | 2.14 | 1.30 |
| Grasp (Abdelfattah et al., 2021) | -0.15 | -0.06 | -0.19 | 0.28 | -0.03 | 1.37 | 1.34 | 1.34 | 2.56 | 1.65 |
| Fisher (Abdelfattah et al., 2021) | 0.03 | 0.08 | 0.20 | -0.79 | -0.12 | 0.97 | 1.04 | 1.05 | 2.10 | 1.29 |
| Plain (Abdelfattah et al., 2021) | 0.12 | 0.22 | 0.26 | -0.39 | 0.05 | 0.90 | 1.01 | 1.03 | 2.10 | 1.26 |
| Synflow (Abdelfattah et al., 2021) | -0.10 | 0.13 | 0.17 | -0.26 | -0.01 | 0.85 | 0.98 | 0.97 | 2.16 | 1.24 |
| NASWOT (Mellor et al., 2021) | -0.45 | 0.15 | 0.32 | -0.42 | -0.10 | 0.92 | 1.00 | 1.02 | 2.03 | 1.24 |
| **Meta-learning** | | | | | | | | | | |
| MetaD2A (Lee et al., 2021a) | 0.63 | 0.51 | 0.40 | 0.26 | 0.45 | 0.01 | 0.01 | 0.01 | 0.01 | 0.01 |
| TANS (Jeong et al., 2021) | 0.43 | 0.61 | -0.15 | -0.17 | 0.18 | 0.01 | 0.01 | 0.01 | 0.01 | 0.01 |
| **DaSS (Ours)** | **0.69** | **0.88** | **0.57** | **0.75** | **0.72** | 0.05 | 0.05 | 0.05 | 0.05 | 0.05 |

Table 4: **End-to-end DaNAS on Unseen Datasets** The distillation performance for the student architecture searched by each method was evaluated on the target validation set. All MACs values shown in the Table are based on $64 \times 64$ pixels and the reported accuracies are the mean and standard deviations over 3 runs.

| Unseen Dataset | | CUB | | Stanford Cars | | DTD | | Quick Draw | |
| | | Acc. (%) | MACs (M) | Acc. (%) | MACs (M) | Acc. (%) | MACs (M) | Acc. (%) | MACs (M) |
|---|---|---|---|---|---|---|---|---|---|
| | Teacher | 37.19 | 1459.51 | 43.25 | 1459.50 | 41.49 | 1456.47 | 44.14 | 1459.50 |
| **Zero-cost proxies** | Grad Norm | $42.23_{\pm 0.25}$ | 1029.07 | $48.07_{\pm 0.08}$ | 1010.42 | $42.80_{\pm 0.22}$ | 1235.98 | $45.30_{\pm 0.10}$ | 981.49 |
| | Snip | $39.55_{\pm 0.05}$ | 1175.61 | $47.83_{\pm 0.21}$ | 1150.48 | $43.19_{\pm 0.49}$ | 1235.98 | $45.24_{\pm 0.06}$ | 981.49 |
| | Grasp | $39.41_{\pm 0.11}$ | 711.06 | $48.22_{\pm 0.12}$ | 719.71 | $44.92_{\pm 0.68}$ | 711.06 | $45.31_{\pm 0.04}$ | 745.29 |
| | Fisher | $39.55_{\pm 0.05}$ | 1175.61 | $48.07_{\pm 0.08}$ | 1010.42 | $43.75_{\pm 0.34}$ | 1076.26 | $45.28_{\pm 0.11}$ | 981.49 |
| | Plain | $40.85_{\pm 0.45}$ | 702.63 | $51.75_{\pm 0.30}$ | 903.76 | $42.98_{\pm 0.68}$ | 1137.60 | $45.24_{\pm 0.10}$ | 981.49 |
| | Synflow | $39.97_{\pm 0.28}$ | 1251.26 | $46.75_{\pm 0.25}$ | 1251.26 | $43.57_{\pm 0.22}$ | 1251.26 | $45.71_{\pm 0.06}$ | 1251.26 |
| | NASWOT | $40.94_{\pm 0.07}$ | 1030.65 | $52.03_{\pm 0.20}$ | 746.44 | $45.47_{\pm 0.51}$ | 874.63 | $45.48_{\pm 0.09}$ | 702.72 |
| **Meta-learning** | MetaD2A | $41.64_{\pm 0.04}$ | 708.75 | $56.74_{\pm 0.36}$ | 734.97 | $45.13_{\pm 0.26}$ | 912.10 | $46.80_{\pm 0.09}$ | 761.73 |
| | TANS | $41.74_{\pm 0.34}$ | 743.99 | $56.00_{\pm 0.12}$ | 995.44 | $45.26_{\pm 0.78}$ | 836.21 | $45.61_{\pm 0.06}$ | 702.60 |
| | **DaSS (Ours)** | $\mathbf{43.48}_{\pm 0.11}$ | 792.97 | $\mathbf{59.09}_{\pm 0.06}$ | 789.13 | $\mathbf{47.86}_{\pm 0.40}$ | 789.13 | $\mathbf{47.27}_{\pm 0.10}$ | 714.13 |

In Table 3, we validate our meta-prediction model with other rapid performance prediction methods. Zero-cost proxies, which are representative rapid performance prediction methods, calculate the scores of architectures by feeding a few batches of the target dataset into randomly initialized architectures. Therefore, they are efficient but cannot consistently predict the distillation performance across all unseen datasets. As shown in Table 3, ours outperforms all rapid NAS baseline models in terms of SRC, including these 6 zero-cost proxies (Abdelfattah et al., 2021; Mellor et al., 2021) and 2 meta-accuracy prediction models (Lee et al., 2021a; Jeong et al., 2021) on 4 unseen datasets. Additionally, the average search time of ours is only $0.05$ GPU seconds, which is competitive with other rapid NAS methods. We further validate the proposed meta-prediction model by conducting end-to-end DaNAS tasks on 4 unseen datasets. For a fair comparison, we randomly sample $3,000$ student architectures and select the Top-1 student architecture whose estimated performance by each method is the highest, then distill the knowledge from the same teacher trained on the target dataset to the selected student architecture with the same number of epochs. The results in Table 4 show that the student architecture found by ours distills the knowledge from the teacher network better than the student architecture found by other baselines on 4 unseen datasets.

## 5.4 Comparison with Meta-accuracy Prediction Models on Unseen Datasets across Domains

We further conducted an evaluation of DaSS on Broader Study of Cross-Domain Few-Shot Learning (BSCD-FSL) benchmark datasets (Guo et al., 2020). The benchmark datasets are from different

Table 5: **End-to-end DaNAS on Unseen Datasets across Domains** We compared DaSS to the meta-accuracy prediction model baselines on BSCD-FSL benchmark containing datasets across domains. All MACs values are based on $64 \times 64$ pixels and the reported accuracies are the mean and standard deviations over 3 runs.

| Unseen Dataset | CropDisease | | EuroSAT | | ISIC | | ChestX | |
|---|---|---|---|---|---|---|---|---|
| | Acc. (%) | MACs (M) | Acc. (%) | MACs (M) | Acc. (%) | MACs (M) | Acc. (%) | MACs (M) |
| Teacher | 99.66 | 1464.66 | 97.41 | 1464.65 | 79.27 | 1464.65 | 40.32 | 1464.65 |
| MetaD2A | $99.68 \pm 0.07$ | 374.10 | $97.21 \pm 0.29$ | 398.06 | $79.27 \pm 1.56$ | 374.10 | $39.95 \pm 1.38$ | 393.50 |
| TANS | $\mathbf{99.72 \pm 0.02}$ | 398.90 | $97.33 \pm 0.48$ | 384.57 | $79.79 \pm 0.89$ | 387.32 | $39.47 \pm 1.40$ | 385.99 |
| **DaSS (Ours)** | $99.71 \pm 0.01$ | 397.84 | $\mathbf{97.95 \pm 0.06}$ | 378.99 | $\mathbf{81.52 \pm 1.08}$ | 374.82 | $\mathbf{42.25 \pm 0.94}$ | 371.47 |

domains compared with the datasets that we have used to learn our meta-prediction model during the meta-training phase. As shown in Table 5, for the EuroSAT, ISIC, and ChestX of the BSCD-FSL benchmark except for the easiest dataset, CropDisease, the student architectures obtained by DaSS are better at KD than other meta-prediction model baselines. The results show that the meta-knowledge of our prediction model with distillation-aware task encoding trained over the meta-training tasks with our adaptation strategy with the teacher is effective in cross-domain datasets as well as in the same-domain datasets.

## 5.5 Validation on a Large-scale dataset with Different Pixels and Models

We validate DaSS with the larger pixels ($256 \times 256$) on ImageNet-1K, a representative large-scale real-world dataset, by conducting end-to-end DaNAS with the pre-trained teacher, which is the largest architecture in a ResNet search space. The search space in this Section is defined by Cai et al. (2020), and it is based on more

Table 6: MACs values in this Table is based on $256 \times 256$ pixels and the architecture configuration is based on Cai et al. (2020).

| | ImageNet-1K ($256 \times 256$) | | | |
|---|---|---|---|---|
| | Acc. (%) | MACs (M) | Depth Config. | Width Config. |
| Teacher | 76.73 | 9843.17 | [4, 4, 6, 4] | [256, 512, 1024, 2048] |
| MetaD2A | 78.43 | 5411.69 | [4, 3, 5, 4] | [208, 512, 816, 2048] |
| TANS | 78.27 | 4211.43 | [2, 3, 4, 2] | [208, 512, 1024, 2048] |
| **DaSS (Ours)** | **79.00** | 4029.64 | [2, 3, 5, 2] | [256, 512, 664, 2048] |

depth and channel widths compared to previous experiments. Using our method DaSS and other meta-accuracy prediction model baselines (MetaD2A and TANS), we select an architecture with a Top-1 score in each method. Then we distill knowledge from the same pre-trained teacher to the obtained student architecture in the same training setting and report the best performance evaluated on the validation set. As shown in Table 6, DaSS successfully searched for more accurate and efficient student architecture, which shows 79.00% accuracy, outperforming the student architectures retrieved by the baseline methods. In sum, the results show that DaSS is not limited to specific models and works well with larger pixels, demonstrating its practicality. More results of experiments with larger pixels are in Appendix F, and more description about the experiment setting is in Appendix E.

## 6 Conclusion

We proposed a novel meta-prediction model, DaSS, that generalizes across datasets, architectures, and teacher networks on DaNAS tasks, which can accurately predict the performance of an architecture when distilling the knowledge of the given teacher network. Existing task-independent DaNAS models require collecting a large number of architecture-accuracy samples as the training set for each dataset-teacher-architecture triplet. The task-conditioned meta-prediction models, on the other hand, can generalize to new datasets, but they are sub-optimal in that they do not consider the KD aspects. In order to devise a meta-prediction model for solving DaNAS tasks, we introduce a novel distillation-aware task encoding based on the embedding of the teacher and a parameter remapping scheme. We also leverage a rapid gradient-based one-shot adaptation of the meta-prediction model on a target task by guiding it with a teacher-accuracy pair, unlike existing meta-accuracy prediction methods, which do not take inner-gradient steps for task adaptation. The experimental results demonstrated that the proposed model outperforms existing rapid NAS methods, such as meta-NAS methods and zero-cost proxies, for the accuracy estimation on the heterogeneous unseen DaNAS tasks.

**Acknowledgements** This work was supported by Institute of Information & communications Technology Planning & Evaluation (IITP) grant funded by the Korea government (MSIT) (No.2019-0-00075, No.2022-0-00713). This material is based upon work supported by Google Research Grant and the Google Cloud Research Credits program with the award (PMF4-4RK0-M9N1-YB37).

**Ethics Statement**   This work has no concerns about the ethical implications.

**Reproducibility Statement**   First, we attached the code https://github.com/CownowAn/DaSS of the proposed method to help with reproducibility. Second, in the appendix document (after references), we enable the experimental results in the main paper to be reproduced by providing a detailed description of the materials and elaborating on the experiment setup, which is organized as follows:

- **Section A** - We provide the details of the *search space* that we used in all experiments in the main document.
- **Section B** - We describe the *implementation details* of functional embedding, architecture embedding, and distillation-aware student architecture embedding as the final input of our model, and meta-surrogate performance prediction model.
- **Section C** - We provide detailed descriptions of *baselines* compared against our mode in the experiments of the main document.
- **Section D** - We elaborate on the detailed *training details*, such as meta-training the proposed prediction model and distilling knowledge from given teacher network to the student architecture, corresponding to the experiments introduced in the main document.
- **Section E** - We provide additional *experimental setup* about time measurements and conditions to sample student architecture in the search space.
- **Section F** - We provide additional *experimental results*.

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

## A  SEARCH SPACE

Our ResNet (He et al., 2016) search space is factorized hierarchical search space; thus, architecture candidates in this search space can have variable depth and channel widths in each stage, which means a sub-module that divides the entire network into smaller units. More specifically, our ResNet search space in Section 5 consists of 4 stages. Similarly to (Cai et al., 2020; Lu et al., 2020), we perform a depth-wise search to select the number of convolutional neural layers for each stage and then perform a channel-wise search to select the number of channel widths for each layer. To further explain the architecture configuration notation in Section 5, **Detph Config.** means how many blocks exist in each stage respectively, and **Width Config.** means the number of channels in the ResNet block convolutional layers at each stage that are obtained by multiplying a channel width shrink ratio by base channel widths predefined for each stage. In Section 5 except for Section 5.5, Depth Config. can be chosen from $\{1, 2, 3, 4\}$ for each stage, the base channel width is $[32, 64, 128, 256]$, and the channel shrink ratio for each layer can be chosen from $\{0.125, 0.25, 0.375, 0.5, 0.625, 0.75, 0.875, 1.0\}$. In the case of Section 5.5, Depth Config. can be chosen from $\{0, 1, 2\}$ and it indicates how much more depth is added to the existing base depth at each stage. Additionally, the base channel width is $[256, 512, 1024, 2048]$, and the channel shrink ratio for each layer can be chosen from $\{0.65, 0.8, 1.0\}$. For all architectures in the search space, the channel shrink ratio of the last layer of each stage is fixed to 1 in order to make the dimension of the output feature the same for each stage.

## B  IMPLMENTATION DETAILS

### B.1  PARAMETER REMAPPING

We consider a network is a composition of $M$ stages, where each $i$-th stage corresponds to $\mathcal{B}_i$. By doing so, we can represent the teacher network and a randomly sampled student architecture in our search space as:

$$\mathbf{t}^\tau(\cdot; \tilde{\boldsymbol{\theta}}^\tau) = \tilde{\mathcal{B}}_M \circ \cdots \circ \tilde{\mathcal{B}}_1(\cdot; \tilde{\boldsymbol{\theta}}_1^\tau), \quad \mathbf{s}^\tau(\cdot; \theta^\tau) = \mathcal{B}_M \circ \cdots \circ \mathcal{B}_1(\cdot; \theta_1^\tau) \tag{8}$$

In our search space, the depth and channel widths of the sampled student architecture $s$ are different from those of the teacher network $\mathbf{t}^\tau(\cdot; \tilde{\boldsymbol{\theta}}^\tau)$. Also, we assume that in the distillation process, the depth and channel widths for each layer in each stage of a given teacher network are always greater than or equal to the student. When performing distillation-aware task encoding, we encode each sampled student architecture in a teacher-adaptive manner with a remapping function, $g$. Since the corresponding architecture configuration of the teacher network and sampled student architecture is different, the remapping process is conducted hierarchically for each stage $m$, by first performing depth-level remapping followed by subsequent width-level remapping. For the sampled student architecture, which has $d$ depth and $k_1, k_2, ..., k_d$ channel width for each depth at stage $m$, remapped student parameters, $\tilde{\boldsymbol{\theta}}_s^\tau = g(\theta^\tau; \tilde{\boldsymbol{\theta}}^\tau)$, is as follows:

$$\tilde{\boldsymbol{\theta}}_{s,m,o}^\tau = \tilde{\boldsymbol{\theta}}_{m,o}^\tau \quad \text{for } o = 1, \ldots, d. \tag{9}$$

$$\tilde{\boldsymbol{\theta}}_{s,m,o,k}^\tau = \tilde{\boldsymbol{\theta}}_{m,o,k}^\tau \quad \text{for } k = 1, \ldots, k_o. \tag{10}$$

### B.2  FUNCTIONAL EMBEDDING

With remapped student candidate, $s(\cdot; \tilde{\boldsymbol{\theta}}_s^\tau)$, it is also important to get a functional embedding that has task-agnostic shape. The spatial dimension of final output logit for the given network varies across tasks and the depth and channel width within the same stage are also different for each sampled student architecture $s$. However, the output dimension of the feature map at each stage is the same across all tasks from the task pool $p(\tau)$, as described in Appendix A. By feeding a fixed Gaussian random input tensor into the remapped student candidate, we can obtain a teacher-conditioned network output feature map, $\mathbf{T} \in \mathbb{R}^{C \times H \times W}$ at the last stage of the student, $\mathcal{B}_M$ while satisfying the condition of getting a task-agnostic shape. The output, $\mathbf{T}$, can be used as the teacher-adaptive functional representation of the corresponding student candidate with respect to the target DaNAS task. By feeding in the teacher-adaptive functional representation tensor, $\mathbf{T}$, into the functional projection layer, $q_f$, we obtain the functional embedding vector, $q_f(\mathbf{T})$. Since we leverage the parameters of a given teacher network, $q_f(\mathbf{T})$ can be teacher-adaptive.

### B.3   ARCHITECTURE EMBEDDING

Our prediction model takes an architecture configuration of sampled student architecture which contains information such as depth and channel widths. As our ResNet search space is composed of 4 stages and 5 layers for each stage, the total number of layers is 20. For each layer, the channel shrink ratio should be chosen from $\{0.125, 0.25, 0.375, 0.5, 0.625, 0.75, 0.875, 1.0\}$ except for Section 5.5. Similar to the one-hot encoding scheme used in (Cai et al., 2020), we can construct one-hot architectural encoding, $\boldsymbol{a}$. Then, by passing the one-hot architecture encoding vector to the architectural projection layer $q_a$, we can get an architecture embedding vector, $q_a(\boldsymbol{a})$. For the case of Section 5.5, we obtain the architecture embedding in a similar way.

### B.4   DISTILLATION-AWARE TASK ENCODING

With $q_a(\boldsymbol{a})$ and $q_f(\mathbf{T})$, which represents architecture embedding and teacher-adaptive functional embedding of sampled student architecture respectively, we concatenate those two embedding vectors and embedding of the teacher and learn another non-linear projection layer, $\sigma$, which outputs distillation-aware network embedding.

## C   BASELINES

In this section, we describe the rapid NAS baselines which we compared with our meta-prediction model. The baseline methods can be divided into two categories, 1) Zero-cost proxies, 2) Meta-prediction model.

### C.1   ZERO-COST PROXIES

Abdelfattah et al. (2021) proposed some metrics based on pruning techniques that can quickly find architectures by leveraging a few batches of target datasets with random state architectures. A metric that measures how a neural network's activations correlate when it is exposed to various inputs over a few batches of the target dataset was also proposed by Mellor et al. (2021). Unlike other methods that require training a sampled architecture at least a few epochs as a proxy task, they have received a lot of attention as very rapid NAS becomes possible.

### C.2   META-PREDICTION MODEL

MAML (Finn et al., 2017) and Meta-SGD (Li et al., 2017) are representative gradient-based meta-learning models to learn to generalize to few-shot regression tasks. These baselines perform set- and distillation-independent prediction on each task. MetaD2A (Lee et al., 2021a) and TANS (Jeong et al., 2021), which are the most closely related to our work, learn dataset-conditioned accuracy prediction models that can generalize to unseen datasets.

## D   TRAINING DETAILS

In this section, we first describe the meta-training database. Then, we describe the details of two main training schemes used in our experiments as follows: 1) Meta-training of the proposed meta-prediction model and 2) Knowledge distillation from the given teacher network to the student architecture.

### D.1   META-TRAINING DATABASE

To learn the proposed meta-prediction model, we collect multiple DaNAS tasks. Each task consists of a dataset and a teacher trained on the dataset, a set of student architectures, and actual accuracies of student architectures after distilling knowledge from the teacher network. As described in Section 5, we use TinyImageNet (Le & Yang, 2015) as the source dataset for meta-training and we generate a heterogeneous set of tasks by class-wisely splitting the dataset into 10 splits. We use subsets consisting of 8 and 2 splits for meta-training and meta-validation, respectively. We train a teacher on each sub-dataset for 180 epochs. Then we randomly sample 2000 student architectures and, for each dataset-teacher network pair, collect 200 student architecture-accuracy pairs after distilling knowledge from the teacher network. For each task, we consider the dataset-teacher pair while randomly selecting distilled student architecture-accuracy pair to train meta-prediction models. Note

that the collection of the meta-training database and meta-training is done **only once** for each search space, as the meta-trained prediction model can be transferred to any DaNAS tasks with different unseen datasets and teacher networks at almost no cost (forward cost to get teacher network accuracy).

## D.2 META-TRAINING THE PROPOSED PREDICTION MODEL

We learn our meta-prediction model over task distribution via episodic training (Vinyals et al., 2016). For each episode, we sample a meta-batch containing multiple tasks for each iteration. More specifically, the number of meta-batch is $8$; in other words, we randomly sample $8$ tasks from the meta-training database for each iteration. Each task consists of a specific teacher network trained on a specific dataset, the accuracy of the teacher network measured on the dataset, 50 student architecture candidates, and their actual accuracy obtained by distilling the knowledge from the teacher network to each student architecture.

For each given task, the meta-prediction model 1) Takes a student architecture and teacher network, 2) Encodes the task in a proposed distillation-aware way; we leverage the embedding of the teacher, architecture embedding of a student, and functional embedding of the student whose parameters are remapped from the teacher, and 3) Predict the distillation performance of the given student architecture. By minimizing the predicted and actual accuracy of the student, we learn the proposed meta-prediction model.

As mentioned in Eq. (6), we utilize the teacher network-accuracy pair for one-shot adaptation. When conducting the inner gradient updates with the teacher-accuracy pair, the total number of inner gradient steps is set to $1$. After the one-shot teacher-guided adaptation, the outer loss is calculated with randomly sampled student architecture-distillation performance pairs for the given teacher network and dataset. During meta-training, we adopt our meta-prediction model with the meta-validation database After fixing our meta-prediction model, a meta-test phase is conducted. As in the meta-learning phase, the one-shot adaptation strategy with the teacher network-accuracy pair was used in the meta-test phase.

## D.3 KNOWLEDGE DISTILLATION

To distill the knowledge of the teacher network to the sampled student architecture, we need to minimize the KD-divergence between the probabilities of the teacher and student network. The probabilities of teacher and student network is as follows:

$$p^T_{t^\tau,i} = \frac{\exp(z_{t^\tau,i}/T)}{\sum_j \exp(z_{t^\tau,j}/T)}, \quad p^T_{s^\tau,i} = \frac{\exp(z_{s^\tau,i}/T)}{\sum_j \exp(z_{s^\tau,j}/T)} \tag{11}$$

where $z_{t^\tau,i}$ and $z_{s^\tau,i}$ are ouputs before the last softmax layer, which is referred as logits, and $T$ is temperature hyper-parameter, which means the degree to soften logits. We use $T$ as 6 for all experiments in ours. With $H$, which is cross-entropy loss, the KD loss $\mathcal{L}_{KD,\alpha}$ is as follows:

$$\mathcal{L}_{KD,\alpha}(p^T_{s^\tau,i}, p^T_{t^\tau,i}; \mathbf{D}^\tau, y) = \alpha \cdot H(y, p^T_{s^\tau,i}) + (1-\alpha) \cdot H(p^T_{s^\tau,i}, p^T_{t^\tau,i}) \tag{12}$$

For every knowledge distillation process, we distill the knowledge of the teacher network to the student by leveraging the KD-aware loss, $\mathcal{L}_{KD,\alpha}$, for $50$ epochs with a learning rate of $5e{-}2$ and we set the value of $\alpha$ as $0.5$.

# E EXPERIMENTAL SETUP

## E.1 EXPERIMENTAL SETUP IN SECTION 5.3

KD is one of the representative model compression techniques to obtain a small network to be used under resource-limited environments or real-time applications. Considering the real-world KD scenarios, it is natural to assume that there is a resource budget in a given environment, and the optimal student architecture must be found within it. In Section 5.3, the teacher network is a network with 7.46M parameters, and MACs constraint on the DaNAS task is as follows: MACs < 1300M when the MACs of the teacher network is about 1450M. All the experimental values shown in this Section are based on $64 \times 64$ pixels.

### E.2 Experimental Setup in Section 5.5

We validate our method with the larger pixels ($256 \times 256$) on ImageNet-1K, a representative large-scale real-world dataset, by conducting end-to-end DaNAS with the largest ResNet defined by Cai et al. (2020) as the teacher in Section 5.5. We first randomly sample $1,000$ architecture candidates; the size of them in terms of MACs is $0.9\times$ smaller than that of the teacher, from the ResNet search space defined by Cai et al. (2020). We predict the performance of the $1,000$ architectures and select the architecture with Top-1 predicted performances computed by each method. Then we distill knowledge from the same pre-trained ResNet50 teacher to the obtained student architecture with the same 50 epochs and $256 \times 256$ pixels and report the best performance evaluated on the validation set.

### E.3 Time Measurement for adaptation and Neural Architecture Search

In Section 5.2, we measure the time for collecting the few-shot student architecture-accuracy pair(s) and the time for conducting the single forward pass to get a teacher network-accuracy pair. In all cases, the time taken was measured on a single NVIDIA's RTX 2080 Ti GPU. As shown in Table 3, we also measure the time required to find an optimal student architecture in DaNAS settings. We randomly sample $3,000$ student architectures and select the Top-1 student architecture found to be most optimal by each method. For every method, after obtaining the total time for scoring for $3,000$ samples first, the average search time per architecture was calculated. As in the previous case, all the time required for searching was measured on a single NVIDIA's RTX 2080 Ti GPU.

## F Validation on Large-Size Pixels

To demonstrate the practical value of the proposed approach, we conducted an experiment on a large-scale real-world dataset, which is ImageNet-1K, with large-size pixels of $256 \times 256$ in Section 5.5. Furthermore, we conducted an experiment similarly on ChestX which is the hardest cross-domain dataset with large-size pixels of $224 \times 224$ in Table 7. As shown in Table 6 and Table 7, our meta-prediction model is robust to various image sizes, increasing practicality. We ob-

Table 7: **Validation on large-size pixels** MACs values in this Table is based on 224 x 224 pixels and the reported accracies are the mean and standard deviations over 3 runs.

| | Acc. (%) | MACs (M) | Depth Config. | Width Config. |
|---|---|---|---|---|
| | **ChestX (224 x 224)** | | | |
| Teacher | 42.00 | 17878.31 | [5, 5, 5, 5] | [32, 64, 128, 256] |
| MetaD2A | 41.13 ± 1.05 | 4795.47 | [2, 4, 2, 2] | [20, 32, 104, 128] |
| TANS | 41.79 ± 0.68 | 4707.01 | [1, 2, 5, 1] | [32, 44, 72, 256] |
| **Ours** | **44.86 ± 0.58** | 4532.10 | [1, 1, 2, 5] | [32, 64, 96, 128] |

served that the proposed method successfully searched for more accurate and efficient student architecture, which shows 79.00% and 44.34% distillation performances on ImageNet-1K and ChestX, respectively, outperforming other meta-accuracy prediction baseline methods.

