# OpenReview forum: "Meta-prediction Model for Distillation-Aware NAS on Unseen Datasets"
_ICLR.cc/2023/Conference — ICLR 2023 notable top 25%_

### Official Review · Reviewer_FiPx · 2022-10-25

**Confidence:** 4
**Correctness:** 3
**Technical Novelty And Significance:** 2
**Empirical Novelty And Significance:** 2
**Recommendation:** 3

**Clarity, Quality, Novelty And Reproducibility:**

Clarity is quite good.

Quality is comparable to recently published related work.

Originality is limited as the proposed predictor mainly adds one element to prior predictors (namely, the teacher network) and employs priorly proposed gradient-based adaptation.


**Strength And Weaknesses:**

## Strengths

**S1.** The method is well-motivated and mostly presented with clarity.

**S2.** The experimental validation suggests the proposed method is effective, as compared to recent methods, in the DaNAS setting.

## Weaknesses

**W1.** My main concern with this submission (and related work) is the set of concerns raised in (Guo et al. ECCV 2020). In particular, the sets of tasks considered in the present submission come mostly from the same domain (e.g., natural images). What would one find out if the present work were to be evaluated across domains like those in (Guo et al. ECCV 2020)?

**W2.** Table 5 shows a large performance gap between the teacher network and the result of DaNAS. What does this gap mean about the performance of the entire pipeline?

**W3.** The experiments are limited to tiny images and the choice of architecture parameters (for teachers and students) were not discussed. In my view, the practical value of the proposed approach is not clear.

**W4.** The remapping of parameters from the teacher network to a given student candidate is not clear to me. In particular, eqs. (6) and (7) do not seem to account for any dimensionality reduction.

## References

Guo et al. A Broader Study of Cross-Domain Few-Shot Learning. ECCV 2020.


**Summary Of The Paper:**

This paper proposes a method for Distillation-aware Network Architecture Search (DaNAS). The main component of the method is a “distillation-aware meta accuracy prediction model” which maps a (student architecture, dataset, teacher network) tuple to a prediction of the accuracy of the student on the particular dataset when the student is trained via knowledge distillation (KD) from the teacher network.

The distillation-aware-meta-accuracy-prediction model is adapted to a target task via gradient-based one-shot adaptation from one teacher-accuracy pair (i.e., the accuracy of the teacher network on the target task).

Experimental results show that the proposed prediction model adapts to target tasks within 3.45 secs on average without direct training on the target tasks. Further, it is shown that the proposed method outperforms existing rapid NAS methods (such as meta-NAS methods and zero-cost proxies) on accuracy estimation and end-to-end DaNAS on a set of unseen tasks.


**Summary Of The Review:**

Similar to related papers, this submission is incremental. However, I am not sure it is practically useful (e.g., tiny images, tiny models -- I think, though this was not clearly specified) and I believe it ignores not-so-recent developments on (the issues of work in) meta-learning.

---

> ### Author Response · Authors · 2022-11-19
> **Response to reviewer FiPx - Part 6 (6/6)**
>
> **W4. The remapping of parameters from the teacher network to a given student candidate is not clear to me. In particular, eqs. (6) and (7) do not seem to account for any dimensionality reduction.**
> * As described in **Section A. Search Space** in the supplementary file, our ResNet search space is composed of 4 stages and a "stage" means a sub-module that divides the entire network into smaller units. And the parameter remapping process is conducted stage-wisely. For each stage, parameters are remapped in consideration of depth and channel widths jointly and hierarchically. If a sampled student architecture has _d_ blocks (depth) and _k_1_, _k_2_, …, _k_d_ channel widths within one stage of attention, first, parameters from the teacher to the _d_-th block are considered. This is what Eq (6) means and it accounts for the dimensionality reduction in depth-level. As mentioned earlier, each block has its own channel widths, so for the _o_-th block (_o_ < _d_), only parameters corresponding to _k_o_ channels are remapped from the teacher’s _o_-th block. Eq (7) is an equation considering dimensionality reduction in channel widths-level.
>
> ***
> To enhance the clarity of the paper, we revised the description of the part remapping of parameters from the teacher network to a given student candidate in the revision. We thank the reviewer's valuable comments.

---

> ### Author Response · Authors · 2022-11-19
> **Response to reviewer FiPx - Part 5 (5/6)**
>
> **W3. The experiments are limited to tiny images and the choice of architecture parameters (for teachers and students) were not discussed. In my view, the practical value of the proposed approach is not clear.**
>
> * To demonstrate the practical value of the proposed approach, we conducted experiments on **large-scale datasets** such as ImageNet-1K with **large-size images of 256x256** and ChestX which is the hardest cross-domain dataset with **large-size images of 224 $\times$ 224** in the below Tables.
> * We included MACs information of architectures to discuss the choice of architectures for teachers and students throughout all experimental results on 9 datasets (CUB, SFC, DTD, STL, CropDisease, EuroSAT, ISIC, ChestX, and ImageNet-1K) with three different image sizes (64 $\times$ 64, 224 $\times$ 224, and 256 $\times$ 256) in all responses. In addition, we describe the detailed architecture configurations obtained by ours and baseline methods in this response on ImageNet-1K and ChestX.
>
> * As shown in the Tables, our meta-prediction model is not only limited to tiny image sizes but also **works well with larger image sizes**, which demonstrates its practicality. We observe that the proposed method successfully searches for more accurate and efficient student architecture with 79.00% and 44.34% accuracies, largely outperforming the student architectures retrieved by the baseline methods (TANS and MetaD2A) on ImageNet-1K and ChestX, respectively.
>
> |           |    ImageNet  (256 $\times$ 256)  |            |               |                       |
> |:---------:|:------------:|:----------:|---------------|-----------------------|
> |           | Accuracy (%) |  MACs (M)  | Depth Config. | Channel Width Config. |
> | _Teacher_ |    _76.73_   | _9843.17_ |  [4, 4, 6, 4] | [256, 512, 1024, 2048] |
> |    TANS   |     78.27    |   4211.43  | [2, 3, 4, 2] | [208, 512, 1024, 2048] |
> |  MetaD2A  |   78.43    |   5411.69  | [4, 3, 5, 4]  | [208, 512, 816, 2048] |
> |  **Ours** |   **79.00**  |   4029.64  | [2, 3, 5, 2]  | [256, 512, 664, 2048] |
>
> |           |    ChestX (224 $\times$ 224)   |            |               |                       |
> |:---------:|:------------:|:----------:|---------------|-----------------------|
> |           | Accuracy (%) |  MACs (M)  | Depth Config. | Channel Width Config. |
> | _Teacher_ |    _42.00_   | _17878.31_ | [5, 5, 5, 5]  | [32, 64, 128, 256]    |
> |    TANS   |     41.59    |   4707.01  | [1, 2, 5, 1]  | [32, 44, 72, 256]     |
> |  MetaD2A  |     41.34    |   4795.47  | [2, 4, 2, 2]  | [20, 32, 104, 128]    |
> |  **Ours** |   **44.34**  |   4532.10  | [1, 1, 2, 5]  | [32, 64, 96, 128]     |
>
>
> * To further explain the architecture configuration notation, as described in **Appendix Section A**, our ResNet search space is factorized into a hierarchical search space and all architecture candidates in this search space are composed of 4 stages. Architecture candidates can have various depths and channel widths at each stage. And the configurations above (_Depth Config._ and _Channel Width Config._) represent the depth and channel widths of the ResNet block convolutional layers at each stage respectively.
>
> ***
> Following the reviewer's suggestion, we included the above experiments on a large-scale dataset (ImageNet-1K) and large-size images (256 $\times$ 256 and 224 $\times$ 224) and discussions about the choice of architectures in the revision (Table 6). We appreciate the reviewer's constructive comments.

---

> ### Author Response · Authors · 2022-11-19
> **Response to reviewer FiPx - Part 4 (4/6)**
>
> **W2. Table 5 shows a large performance gap between the teacher network and the result of DaNAS. What does this gap mean about the performance of the entire pipeline?**
> * KD is one of representative model compression techniques to obtain small network to be used under resource limited environment or real-time applications. Considering the real-world KD scenarios, it is natural to assume that there is a resource budget in a given environment, and the optimal student architecture must be found within it. Thus, we performed DaNAS under a resource-limited environment, which could lead to a performance gap with the teacher network. More specifically, for the experiment in Table 5 (the previous version of the paper), the MACs of the student architecture were limited to less than 20% of the MACs of the teacher network. Therefore, there could be a performance gap between the student network and the teacher network.
> As you pointed out in your summary of the review that we need to use larger models to increase practicality, we additionally conducted End-to-end DaNAS for the larger teacher network with _7.46M_ Params. (previous teacher network is a network with _5.89M_ Params.), assuming a scenario with a looser MACs constraint (MACs < 1300M, when the MACs of the teacher network is about 1450M). As can be seen from the Table results, with a looser MACs constraint, there is not a large gap between the teacher network and student networks, and even student networks outperform the teacher network. Even in this scenario, we can see that the student architecture found by ours distills the knowledge from the teacher network better than the student architecture found by other baselines for the CUB, SFC, DTD datasets. In the case of the STL dataset, student architecture found by ours shows competitive results compared to TANS. All MACs values shown in the Table are based on 64x64 image size.
> | Unseen Dataset |      CUB     |           |      SFC     |           |      DTD     |           |      STL     |           |
> |:--------------:|:------------:|:---------:|:------------:|:---------:|:------------:|:---------:|:------------:|:---------:|
> |                | Accuracy (%) |  MACs (M) | Accuracy (%) |  MACs (M) | Accuracy (%) |  MACs (M) | Accuracy (%) |  MACs (M) |
> |    _Teacher_   |    _37.19_   | _1459.51_ |    _43.25_   | _1459.50_ |    _41.49_   | _1456.47_ |    _76.29_   | _1459.46_ |
> |    Grad Norm   |     42.19    |  1029.07  |     48.05    |  1010.42  |     42.83    |  1235.98  |     78.93    |  1175.61  |
> |      Snip      |     39.60    |  1175.61  |     48.04    |  1150.48  |     42.83    |  1235.98  |     79.50    |  1125.02  |
> |      Grasp     |     39.29    |   711.06  |     48.09    |   719.71  |     45.64    |   711.06  |     79.43    |   785.43  |
> |     Fisher     |     39.60    |  1175.61  |     48.05    |  1010.42  |     43.48    |  1076.26  |     78.93    |  1175.61  |
> |      Plain     |     40.43    |   702.63  |     52.10    |   903.76  |     43.37    |  1137.60  |     80.09    |   716.38  |
> |     Synflow    |     40.23    |  1251.26  |     46.98    |  1251.26  |     43.37    |  1251.26  |     78.96    |   700.22  |
> |     NASWOT     |     40.87    |  1030.65  |     51.89    |   746.44  |     45.74    |   874.63  |     80.10    |   736.74  |
> |      TANS      |     41.61    |   743.99  |     55.88    |   995.44  |     44.40    |   836.21  |   **80.24**  |   726.37  |
> |     MetaD2A    |     41.67    |   708.75  |     56.33    |   734.97  |     45.26    |   912.10  |     79.28    |   752.26  |
> |    **Ours**    |   **44.24**  |   747.19  |   **59.65**  |   735.34  |   **47.04**  |   741.57  |     80.16    |   735.34  |
>
> * About the architecture found in the absence of extreme MACs constraint, we compared the performance of the same architecture with different training pipelines, in case you wonder why the student architecture, which is a smaller network than the teacher, has higher performance (DTD dataset, the student architecture found by 'ours'). As shown in the below Table, in the case of a student network smaller than the teacher network, the performance is worse than that of the teacher when scratch training is performed, but it is even better when knowledge distillation is performed.
> |           |           DTD          |                  |
> |:---------:|:----------------------:|:----------------:|
> |           | Knowledge Distillation | Scratch Training |
> | _Teacher_ |         _41.49_        |         -        |
> |    Ours   |          47.04         |       38.09      |
>
> ***
> We appreciate the reviewer's helpful comments and updated the revision by adding detailed descriptions of experimental scenarios and including the above experiments conducted with larger teacher and student architectures (Table 4).

---

> ### Author Response · Authors · 2022-11-19
> **Response to reviewer FiPx - Part 3 (3/6)**
>
> **W1. My main concern with this submission (and related work) is the set of concerns raised in (Guo et al. ECCV 2020). In particular, the sets of tasks considered in the present submission come mostly from the same domain (e.g., natural images). What would one find out if the present work were to be evaluated across domains like those in (Guo et al. ECCV 2020)?**
> * Please note that we successfully validated our method on the DTD dataset in the paper, which is a representative **out-of-distribution dataset** consisting of texture images (non-natural images).
> * We further conducted an evaluation on **BSCD-FSL (Broader Study of Cross-Domain Few-Shot Learning) benchmark containing datasets across multiple domains**,  which do not come from the same domain (natural images), by referring to the related work (Guo et al. ECCV 2020) that the reviewer suggested. Specifically, we validated our model on new target datasets such as CropDisease, EuroAST, ISIC, and ChestX, which have different domains compared with the datasets that we have used to learn our model during meta-learning as below:
> |           |  CropDisease |           |    EuroSAT   |            |     ISIC     |            |    ChestX    |            |
> |:---------:|:------------:|:---------:|:------------:|:----------:|:------------:|:----------:|:------------:|:----------:|
> |           | Accuracy (%) |  MACs (M) | Accuracy (%) |  MACs (M)  | Accuracy (%) |  MACs (M)  | Accuracy (%) |  MACs (M)  |
> | _Teacher_ |    _99.66_   | _1464.66_ |    _97.41_   |  _1464.65_ |    _79.27_   |  _1464.65_ |    _40.32_   |  _1464.65_ |
> |    TANS   |     **99.74**    |   398.90  |     97.26    |   384.57   |     80.31    |   387.32   |     37.85    |   385.99   |
> |  MetaD2A  |     99.61    |   374.10  |     97.30    |   398.06   |     77.72    |   374.10   |     38.37    |   393.50   |
> |  **Ours** |     99.72    |   397.84  |   **98.00**  | 378.99 |   **82.38**  | 374.82 |   **41.83**  | 371.47 |
>
> * As can be seen from the table results, for the EuroSAT, ISIC, and ChestX of the BSCD-FSL benchmark except the most easy dataset, CropDisease, **the student architectures obtained by ours have superior performances** to other meta-prediction baselines. The results show that the meta-knowledge of our prediction model with distillation-aware encoding trained over the meta-training tasks with our adaptation strategy with teacher network is effective in cross-domain datasets as well as in the same-domain datasets.
>
> ***
> We thank the reviewer for the constructive comments and included the above cross-domain experimental results in the revised version of the paper (Table 5).

---

> ### Author Response · Authors · 2022-11-19
> **Response to reviewer FiPx - Part 2 (2/6)**
>
> **Originality is limited as the proposed predictor mainly adds one element to prior predictors (namely, the teacher network) and employs priorly proposed gradient-based adaptation.**
> * We believe that the reviewer overlooked the difficulty of the DaNAS problem as including the teacher network into a meta-predictor is a non-trivial, challenging task.
>
> | Training Pipeline      |  Teacher | Arch. 1 Acc. (%) | Arch. 2 Acc. (%) |            Comparison           |
> |------------------------|:--------:|:---------------------------:|:---------------------------:|:-------------------------------:|
> | Scratch Training       |     -    |            42.97            |            46.84            | Arch. 1 < Arch. 2 |
> | Knowledge Distillation | ResNet34 |            49.53            |            47.63            | Arch. 1 > Arch. 2 |
>
> * In DaNAS domain perspective,
>
>   * The previous meta-learning-based accuracy predictors designed for the NAS problem are not suitable to for DaNAS task and thus we needed to design a novel meta-predictor for DaNAS task. To validate this motivation, we conducted the following experiment on the subset of TinyImageNet. As shown in the above Table, the suitable architectures for scratch training and KD training are different. However, while the teacher model largely affect the final performance of the architecture that is trained with knowledge distillation, existing performance predictors do not consider the teacher model at all.
>
>    * How to design the predictor to consider the distillation process is an important and non-trivial problem for DaNAS. We tackle this problem by **not simply adding one element to prior predictors designed for only considering the "dataset" and ignoring components of knowledge distillation such as the teacher network trained on the target dataset, but also by proposing a novel distillation-aware task encoding**. Such distillation-aware task encoding is based on the **parameter remapping** from a specific teacher to a student and the **functional embedding** of the remapped student.
>
>   * In addition, we propose a **gradient-based one-shot adaptation scheme** with the teacher network and its accuracy pair. The idea of the pre-trained **teacher** network-its accuracy as an adaptation ingredient is simple yet effective for DaNAS tasks as it enables rapid architecture search. The DaNAS environment is different from the conventional NAS in that we already have a trained model, the teacher. There is no need to wait to get the accuracy-architecture pair(s) for adaptation since we can get the accuracy of the teacher with a single forward pass.
>
>    * Furthermore, ours is the **first** work to tackle the generalization issue on various datasets including the large-scale dataset and datasets from the out-of-distribution domain while most DaNAS works have validated their frameworks on a very limited number of datasets such as ImageNet-1K, Cifar-10, and Cifar-100.
>
>    * Ours is the **first** lightweight and rapid search method on target datasets while most DaNAS methods are extremely costly to be used for multiple datasets.
>
> * In Meta-NAS perspective,
>    * We are the **first** to validate the effectiveness of the **gradient-based adaptation** on **Meta-NAS** problem, although we borrow the gradient-based adaptation from the existing meta-learning methods, they are limited to few-shot classification task and artificial regression task so that simply using them to Meta-(Da)NAS problem without elaborately designed predictors work poorly. The existing Meta-NAS methods such as MetaD2A and TANS have not leveraged the power of gradient-based adaptation on NAS task.

---

> ### Author Response · Authors · 2022-11-19
> **Response to reviewer FiPx - Part 1 (1/6)**
>
> **Similar to related papers, this submission is incremental. However, I am not sure it is practically useful (e.g., tiny images, tiny models -- I think, though this was not clearly specified)**
>
> * During the rebuttal period, we significantly **enhanced the practical value** of the proposed method by demonstrating it on various scenarios as the reviewer FiPx suggested as follows:
>   * **Datasets in different domains** We experimented on 9 datasets, including fine-grained datasets (CUB, SFC, and CropDisease), out-of-distribution datasets (DTD - texture images, EuroSAT - satellite images, ISIC - medical images about skin lesions, ChestX - X-ray images). This shows that our method works well on datasets that are very different from the datasets used in the meta-learning phase.
>   * **Large images** We added the end-to-end DaNAS experimental results conducted on datasets consisting of large images with 224 $\times$ 224 pixels, and 256 $\times$ 256 pixels.
>
>   * **Large-scale datasets** We conducted end-to-end DaNAS experiments on ImageNet-1K which is one of the representative large-scale datasets and clearly outperformed other baseline models in the response.
>
>   * **Larger models** We experimented with larger teachers network with more parameters. The results show that our method is not limited to tiny models and is robust to the various model sizes.
>
> **I believe it ignores not-so-recent developments on (the issues of work in) meta-learning.**
>
> * We believe that the reviewer is confused between the **meta-learning on few-shot classification** task and **Meta-NAS** problem.
> * Meta-NAS problem is a **large-scale meta-learning problem, which requires to meta-learn over a large number of datasets and architectures**, and thus training them with existing meta-learning methods is **infeasible or impractical, as they target few-shot learning** tasks and do not scale to training with large number of data instances.
> * This is why existing Meta-NAS methods such as MetaD2A (ICLR 2021) and TANS (NeurIPS 2021) focus on constructing embeddings of the dataset and architectures, and meta-learn the predictor using those task embeddings, rather than trying to benefit from the recent developments in meta-learning.
> * In addition, naively borrowing existing meta-learning methods on NAS task without **carefully designing a predictor** will fail on our DaNAS task, as shown from the poor results of MAML and Meta-SGD on DaNAS tasks of our paper.
> * Note that we compare our method with the state-of-the-art Meta-NAS methods, such as MetaD2A (ICLR 2021) and TANS (NeurIPS 2021).

---

> ### Author Response · Authors · 2022-11-29
> **A Gentle Reminder**
>
> Dear reviewer FiPx,
>
> We sincerely hope that you go over our detailed responses and the revision, which we believe addresses all your concerns and contains all additional experimental results requested by you. To help you quickly find what we did during the rebuttal period and further save more of your time, we briefly summarize our response below:
>
> ***
> - **Part 1 (1/6)**:
>   - Brief summary of additional experiments we conducted to address your concerns.
>   - The difference between "meta-learning on few-shot classification task" and "Meta-NAS problem" in addition to our contribution.
> ***
> - **Part 2 (2/6)** :
>   - Our novelty from the perspective of DaNAS domain.
>   - Our novelty from the perspective of Meta-NAS.
> ***
> - **Part 3 (3/6)** :
>   - Experiments on the BSCD-FSL (Broader Study of Cross-Domain Few-Shot Learning) benchmark containing datasets across multiple domains to address your concern.
> ***
> - **Part 4 (4/6)** :
>   - Experiments with larger teachers and students with more parameters to address your concern.
> ***
> - **Part 5 (5/6)** :
>   - Experiments on datasets consisting of large images to address your concern.
>   - Experiments on large-scale datasets such as ImageNet-1K.
> ***
> - **Part 6 (6/6)** :
>   - Additional description of parameter remapping to address your concern.
> ***
>
> We appreciate your insightful and constructive comments once again, and believe that they helped significantly strengthen our paper. We sincerely thank you for your time and efforts in reviewing our paper, and hope you reconsider your rating based on the rebuttal. Please let us know if you have any further questions or concerns.
>
> Best regard, The authors

---

> > ### Comment · Reviewer_FiPx · 2022-11-30
> > **Re: A Gentle Reminder**
> >
> > Hi authors,
> >
> > I appreciate your detailed replies and extensive revisions. After looking at your summary of revisions I would imagine your paper could be above the acceptance threshold -- though I have not reviewed the paper anew. I am unsure at the moment how ICLR will want to handle such extensive revision of a submission.
> >
> > Please see the FAQ at https://iclr.cc/Conferences/2023/ReviewerGuide, and the following answer in particular:
> >
> > "New experiments should not significantly change the content of the submission. Rather, they should be limited in scope and serve to more thoroughly validate existing results from the submission."
> >
> > Re meta-learning, I still have concerns there -- though your paper could be above the acceptance threshold besides these concerns. The concern with meta-learning, even as applied in your method, is the lack of performance guarantees when the assumption of domain similarity does not hold.
> >
> > Best,
> >
> > Reviewer FiPx

---

> > > ### Author Response · Authors · 2022-12-01
> > > **Thank you for your feedback.**
> > >
> > > Dear Reviewer FiPx,
> > >
> > > We sincerely appreciate your reply. We do not believe that the additional results we provided will make the paper fall into the criterion you mentioned, since we did **not change our core method and conclusion**. We faithfully conducted experiments and revised the papers to address your concerns rather than to propose something new, and those new results **further strengthened our main argument and conclusion** rather than going against them.
> > >
> > > Also, as clearly described below, after checking the question and answer in the FAQ you mentioned (at https://iclr.cc/Conferences/2023/ReviewerGuide),
> > >
> > > - **Q**: Am I allowed to ask for additional experiments?
> > > - **A**: You can ask for additional experiments. New experiments should not significantly change the content of the submission. Rather, they should be limited in scope and serve to more thoroughly validate existing results from the submission.
> > >
> > > the question and answer in this **"FAQ for Reviewers"** serve as a guide for **reviewers** when they request additional experiments. We believed that your comments were very worthy of consideration within the scope of our paper, and the results of the experiments suggested by you support our method consistently with our existing results. Therefore, as we mentioned before, there seems to be nothing to be unsure of.
> > >
> > > Additionally, we believe that the purpose of the interactive discussion period for openreview-style review process is to promote active communication among the reviewers and authors to further strengthen the paper **in a more constructive manner through the discussion**. Thanks to your constructive comments, we believe that we have strengthened the paper by including more results that further verify the efficacy of our methods, which further strengthens our work.
> > >
> > > Since we have addressed all of your concerns, we politely ask you reflect your thought that our paper is above the acceptance threshold on your rating.
> > >
> > > Also, we **verified our method on the cross-domain datasets** by covering all datasets in the paper [1] you referred, and reported that our method achieves performance improvements on the cross-domain datasets. Note that their domains are highly dissimilar from the datasets we used during the meta-training. The experimental results of this are in **Part 3 (3/6)** of our response in more detail.
> > > The below is the complete list of the datasets covered in our experiments, including the cross-domain datasets:
> > >
> > >   - natural-image domain datasets:
> > >     - STL10 (STL)
> > >     -  CropDisease - agriculture images
> > >   - **fine-grained** datasets:
> > >     - CUB
> > >     - Stanford Cars (SFC)
> > >   - **out-of-distribution** datasets:
> > >     - DTD - texture images
> > >     - EuroSAT - satellite images
> > >     - ISIC - gray medical images about skin lesions
> > >     - ChestX - X-ray images
> > >   - **large-scale real-world** dataset:
> > >     - ImageNet-1K
> > >
> > > **We experimented on 9 datasets, with a more wide variety of ranges than [1] and only one dataset short of the experiments done in paper [2] which experimented on 10 datasets**. We believe that this is more than sufficient to verify the efficacy of our method, and that the results **empirically verify how our method does on datasets that are dissimilar from the datasets in meta-training set**.
> > >
> > > As for the performance guarantee based on the similarity across the tasks, we empirically verified that our method achieves good performance on cross-domain tasks with large disparity among the meta-training and meta-test task, in **Part 3 of our response**. Further, we believe that applying Bayesian meta-learning methods such as Bayesian TAML [3] may allow us to model the uncertainty for unseen tasks to further tackle out-of-domain tasks and this could be a potential future research direction. Thank you for your insightful suggestion. However, as shown in the paper, meta-testing on tasks that differ too much from the set of training tasks will simply reduce to standard learning on the new tasks while discarding all meta-knowledge, and the basic assumption of meta-learning is that there is certain relatedness among the tasks we see during meta-training and meta-test time, since meta-learning aims to generalize over a distribution of tasks. Addressing this is beyond the scope of our paper, as this is a more fundamental challenge with meta-learning.
> > >
> > > Thank you for your time and efforts in reviewing our paper once again and we sincerely appreciate your constructive comments which have contributed greatly to strengthening the effectiveness of our method.
> > >
> > > Best regard, The authors
> > > ***
> > > **References**
> > >
> > >   [1] A Broader Study of Cross-Domain Few-Shot Learning, ECCV 2020.
> > >
> > >   [2] Meta-Dataset: A Dataset of Datasets for Learning to Learn from Few Examples, ICLR 2020
> > >
> > >   [3] Learning to Balance: Bayesian Meta-Learning for Imbalanced and Out-of-distribution Tasks, ICLR 2020

---

> > > ### Author Response · Authors · 2022-12-03
> > > **A Gentle Reminder - The end of the discussion phase is approaching**
> > >
> > > Dear FiPx,
> > >
> > >
> > > We sincerely hope that you review our response and the revision since we now have **less than 10 days** to have interactive discussions.
> > > We have faithfully addressed your concern and explained what you were unsure about the ReviewerGuide (reply right below).
> > >
> > > A summary of what we did during the rebuttal period is in **Part 1**, and in particular, addressing concerns about meta-learning is in **Part 3**.
> > >
> > > Thank you for your time and efforts in reviewing our paper once again and if you have any additional questions or concerns, please let us know.
> > >
> > >
> > > Best regard, The authors

---

> > > ### Author Response · Authors · 2022-12-05
> > > **A Gentle Reminder - The interactive discussion phase ending in less than a week**
> > >
> > > Dear Reviewer FiPx,
> > >
> > >
> > > Could you review the detailed responses and the revision we provided during the rebuttal period in addition to the explanation of the uncertainties you had about the ReviewerGuide?
> > >
> > > We have conducted additional experiments based on your constructive comments and sincerely hope to receive your feedback on them.
> > >
> > > But unfortunately, so far we haven't received your feedback on them, and the discussion phase ends in less than a week.
> > >
> > > Therefore, we respectfully hope you review what we have done and leave feedback once again.
> > >
> > > Also, please let us know of any other things we should clarify or address since we have only less than a week before the end of the interactive discussion phase.
> > >
> > > We appreciate your helpful comments which we believe have largely improved our paper and thank you for spending your precious time on our paper.
> > >
> > > Best regard, The authors

---

### Official Review · Reviewer_L5tN · 2022-10-25

**Confidence:** 4
**Correctness:** 3
**Technical Novelty And Significance:** 3
**Empirical Novelty And Significance:** 3
**Recommendation:** 8

**Clarity, Quality, Novelty And Reproducibility:**

The paper is implementable, I think the section on encoding could be clearer (this is the size of embedding for teacher, dataset - hzt and size of encoding of student - ht). Additionally its good that the code is included to reproduce results.

**Strength And Weaknesses:**

Strength
1. Paper proposes a new method for fast generation of student architectures when training on new problems of knowledge distillation

Weaknesses:
1. Severely restricted architecture space: the NAS template restricts students to be smaller than teacher, further the students either have no kth layer or just copy the parameters of teacher for kth layer. While this is good to demonstrate empirical results, it will severely limit the impact in real world problems.
2. the paper only shows results on small image datasets, I would like to see if model does well on other non image modalities and other real world image datasets - imagenet/faces
3. I am not sure how the paper prevents the model from learning 1-1 mapping - i.e. every "optimal" student is same architecture as teacher. It needs to be clearly exposition as the teacher parameters are simply being copied.
4. The idea shows good results but highly constrained - I think at least exploratory experiments showing use of LSTM etc will greately strengthen the paper

**Summary Of The Paper:**

NAS for knowledge distillation requires architecture search for a student network which will learn well using teacher network. The method DaNAS proposes learning meta model for accuracy prediction given a dataset, teacher and student architecture and using this to find optimal student network.

**Summary Of The Review:**

Authors describe an algorithm to generate student architectures for distillation from teacher network and dataset. The show emprically promising results.

---

> ### Author Response · Authors · 2022-11-19
> **Response to reviewer L5tN - Part 3 (3/3)**
>
> **W3. I am not sure how the paper prevents the model from learning 1-1 mapping - i.e. every "optimal" student is same architecture as teacher. It needs to be clearly exposition as the teacher parameters are simply being copied.**
> * As mentioned in **Section 5.3. Comparison with Rapid NAS Methods**, we randomly sample 10,000 student architectures except for the teacher network for a fair comparison and select the top-1 student architecture whose estimated performance is the highest.
> Furthermore, considering the real-world scenario, we assumed a resource budget exists in a given environment, and the NAS methods should find the optimal student architecture within it. We performed additional distillation-aware NAS under a resource-limited environment.
>
>
> **W4. The idea shows good results but highly constrained - I think at least exploratory experiments showing use of LSTM etc will greatly strengthen the paper**
> * We sincerely appreciate your suggestions and insightful comments. In order to increase the flexibility of our method, we plan to proceed with validation in other modalities during the remaining rebuttal period or until camera-ready, depending on whether or not our submission is accepted, and we expect our paper to be strengthened.

---

> > ### Comment · Reviewer_L5tN · 2022-11-29
> > **Response to authors**
> >
> > I appreciate the responses - and clarification for network architecture reduction, combined with new experiments. The two major questions that still remain for me are
> > 1. Application for LSTM - other architectures for neural networks
> > 2. Application for non image datasets.
> > Without those two - I am inclined to keep my rating as is. I do think the rating is fair representation of both the impact/contribution of paper and the work done by authors.

---

> ### Author Response · Authors · 2022-11-19
> **Response to reviewer L5tN - Part 2 (2/3)**
>
> **W2. The paper only shows results on small image datasets, I would like to see if model does well on other non image modalities and other real world image datasets - imagenet/faces.**
> * Following your comments, we validated our method on a **real-world image dataset (ImageNet-1K)** as well as small image datasets by conducting end-to-end DaNAS to obtain an efficient student architecture for the largest ResNet which is defined by [4] and trained on ImageNet-1K as the teacher. We first randomly sample 1000 architecture candidates, which the size of them in terms of MACs is 0.9 $\times$ smaller than the teacher, from the ResNet search space defined by [4]. Using our meta-learned architecture performance predictors, TANS and MetaD2A, we predict the final performance of the 1000 architectures and select architectures with top-1 predicted performances computed by predictors as the obtained student architectures, respectively. Then we distill knowledge from the same pre-trained teacher to the obtained student architecture with the same 50 epochs and the same 256 $\times$ 256 image size and report the best performance evaluated on the validation set as below:
>
> |           |    ImageNet    |            |               |                       |
> |:---------:|:------------:|:----------:|---------------|-----------------------|
> |           | Accuracy (%) |  MACs (M)  | Depth Config. | Channel Width Config. |
> | _Teacher_ |    _76.73_   | _9843.17_ |  [4, 4, 6, 4] | [256, 512, 1024, 2048] |
> |    TANS   |     78.27    |   4211.43  | [2, 3, 4, 2] | [208, 512, 1024, 2048] |
> |  MetaD2A  |   78.43    |   5411.69  | [4, 3, 5, 4]  | [208, 512, 816, 2048] |
> |  **Ours** |   **79.00**  |   4029.64  | [2, 3, 5, 2]  | [256, 512, 664, 2048] |
>
> * We observed that the proposed method successfully searched for more accurate and efficient student architecture, which shows 79% accuracy compared with other baseline methods (TANS and MetaD2A) on the real-world dataset - ImageNet-1K. By the regularization effect of the KD, the student obtained by ours outperformed even the teacher.
>
> * ResNet search space defined by [4]:
> * To further explain the architecture configuration notation, as described in **Appendix Section A**, our ResNet search space is factorized hierarchical search space, and all architecture candidates in this search space are composed of 4 stages. Architecture candidates can have various depths and channel widths at each stage. Moreover, the configurations in the Table above (_Depth Config._ and _Channel Width Config._) represent the depth and channel widths of the ResNet block convolutional layers at each stage, respectively. A more detailed explanation is below:
>    * _'Depth Config'_ can be chosen from {0, 1, 2} and it indicates how much more depth is added to the existing base depth [2, 2, 4, 2] at each stage.
>   * _'Channel Width Config.'_ means the number of channels in the ResNet block convolutional layers at each stage that are obtained by multiplying a channel width shrink ratio ({0.65, 0.8, 1.0}) by base channel widths predefined for each stage, which is [256, 512, 1024, 2048] in this case.
>
> * Furthermore, by following the Reviewer FiPx’s suggestion, we conducted the experiments to validate our model on **more challenging datasets in cross domains** such as CropDisease, EuroSAT, ISIC, and ChestX datasets that contain plant disease images, satellite images, dermoscopic images of skin lesions, and X-ray images, respectively. Such datasets have varying dissimilarity from natural images used in the meta-training phase according to the 3 orthogonal criteria [5]: 1) the existence of perspective distortion, 2) the semantic content, and 3) color depth. Please refer to "Response to reviewer FiPx - Part 2" to see the results. The student architecture obtained by ours shows competitive performance (CropDisease) and clearly outperformed performances (EuroSAT, ISIC, and ChestX).
> ***
> #### **References**
>
> [4] Once for All: Train One Network and Specialize it for Efficient Deployment, ICLR 2020.
>
> [5] A Broader Study of Cross-Domain Few-Shot Learning, ECCV 2020.
>
> ***
> Following the reviewer's suggestion, we included the above experiments on a large-scale dataset (ImageNet-1K) in the revision (Table 6). We appreciate the reviewer's constructive comments.

---

> ### Author Response · Authors · 2022-11-19
> **Response to reviewer L5tN - Part 1 (1/3)**
>
> **I think the section on encoding could be clearer (this is the size of embedding for teacher, dataset - hzt and size of encoding of student - ht).**
> * We appreciate your constructive comments. Following your comments, we update the description of the encoding part in **Section B. Implementation Details** in the supplementary file. Please refer to it.
>
> **W1. Severely restricted architecture space: the NAS template restricts students to be smaller than teacher, further the students either have no kth layer or just copy the parameters of teacher for kth layer. While this is good to demonstrate empirical results, it will severely limit the impact in real world problems.**
> * This is the reviewer’s misunderstanding because we follow the typical KD scenario rather than limiting the architecture space due to the parameter remapping. KD is one of the representative model compression techniques to get a computational efficiency model by transferring knowledge from the “larger” teacher model to the “smaller” student model. Thus, the assumption that the search for a student architecture that is smaller than the teacher model in the KD scenario is reasonable and natural.
> Furthermore, we have utilized the parameter remapping technique, which copies parameters from the front layers of the source model with the same number of layers of the target model and pastes them to the target model. Such parameter remapping technique is one of the representative parameter remapping techniques that many existing NAS works [1,2,3] have applied to their method and demonstrated its effectiveness of it during the traditional search process, even if they have not considered KD and meta-learning framework. Finally, we provided experiments to validate our method on more practical scenarios, such as **large-scale real-world datasets** in the below answers (W2) and **more challenging datasets in cross domains** (Response to reviewer FiPx - Part 3). Thus, we highly believe that our problem setting related to the size of the teacher and student and the parameter remapping technique we choose are reasonable and practical enough.
> ***
> #### **References**
>
> [1] Path-Level Network Transformation for Efficient Architecture Search, ICLR 2018
>
> [2] Fast Neural Network Adaptation via Parameter Remapping and Architecture Search, ICLR2020
>
> [3] FNA++: Fast Network Adaptation via Parameter Remapping and Architecture Search, TPAMI 2020

---

> ### Author Response · Authors · 2022-11-29
> **Gentle Reminder - Dear Reviewer L5tN (11/28)**
>
> Dear reviewer,
>
> Could you check our responses to your comments as well as the revision that reflects them?
> As you suggested, we included the experiments on a large-scale dataset, ImageNet-1K, in the revision (Table 6). In addition, we clarified the descriptions for the encoding and parameter remapping parts in the introduction, the method, and Section B of the appendix. We would like to hear your feedback since the end of the discussion phase is less than two weeks.
>
> Thanks, Authors

---

### Official Review · Reviewer_T2PL · 2022-10-28

**Confidence:** 4
**Correctness:** 3
**Technical Novelty And Significance:** 3
**Empirical Novelty And Significance:** 4
**Recommendation:** 8

**Clarity, Quality, Novelty And Reproducibility:**

Some comments on the clarity:
- typo: many sentences are too difficult to understand, eg we propose a distillation-aware task encoding function that remaps parameters from the teacher to the student and considers the output from the embedding network of the remapped student as the embedding to estimate the teacher network’s impact on the final accuracy of the distilled model.
- The writing is really hard to follow, expecially when combing a lot of new concepts.

**Strength And Weaknesses:**

Strength:

S1: The idea is quite novel. The previous predictor-based NAS methods focus on accuracy, latency, FLOPs, and #params. The prediction of performance for a student after KD considering the given teacher and task is novel.

S2:The motivation is interesting: previous DaNAS can not generalize well to any other tasks with new combinations and teachers(not flexible), thus requiring retraining for these tasks and teachers' combination. Meta NAS enhance the fast adaption to the new task with less training cost, but none of them considers the influence of the KD. This paper proposes a method that tackles the above two obstacles: meta leaned prediction model which is 1) meta-trained (for fast adaption); 2) takes as input the teacher encoding and is guided by the accuracy of the teacher during the mete training.

However, I still have below concerns

W1.Some logic is hard to follow, for example, in Intro, conventional NAS is sub-optimal for searching students under KD because the optimal student distilled from the teacher is different from an architecture trained from scratch. I do not understand why the training pipeline difference would cause conventional NAS not work. Maybe more sentences can help like: if we want to adopt NAS to search for a suitable architecture for KD, we should consider the influence of the components from KD (teacher and source dataset, etc.). But traditional NAS is designed for searching an architecture according to its evaluations trained from scratch instead of the KD aspects.
The logic should be explained. And there are many other examples, in the abstract, the author claims that meta-learning NAS is sub-optimal for DaNAS. But from my perspective, they are designed originally for solving the generalization issue for NAS, and of course, they are not the optimal choice for DaANS. The author claims they are not optimal  just because they want to adopt meta-learning in DaNAS and find that they do not consider teacher apsect so they are not optimal. The logic needs to be reconsidered here and for the rest of the paper.

W2. typo: many sentences are too difficult to understand, eg on page 2: we propose a distillation-aware task encoding function that remaps parameters from the teacher to the student and considers the output from the embedding network of the remapped student as the embedding to estimate the teacher network’s impact on the final accuracy of the distilled model.

W2.1. The writing is really hard to follow, especially when combing a lot of new concepts. More, concepts need to be expiated clearly for example the setting of the few-shot learning under this paper's scenario.

W3.The layout is so confusing that the reference is several pages away from the table/figure and the caption is not clear to understand. For example, the main results are in Tab2, what is the unit of the results, I can not figure out them only by a few dataset names. And what are the model components: Set. PR. and Gui.? Please explain more in the caption (Tab2,3,4,5).  The figure is also confusing, what is collecting time in figure 4?

W4.Please explain more clearly in section 5 about your training and searching setting when combining Meta, KD, few-shot learning and one-shot adaption, and NAS. This is really confusing.


**Summary Of The Paper:**

This paper propose a teacher awared accuracy predictior which solves the issue for DaNAS that previous predictors are not aware of influence of the teacher model. The results demonstrate the efficacy of the proposed methods.

**Summary Of The Review:**

See previous sections for detailed comments, and I recommend the authors to revise the paper accordingly.


# Updates after reading the response

I thank the authors for providing me a informative response that addresses nearly all of my previous concerns.  As this work proposes to combine two sub-domain of NAS, distillation aware NAS and meta-learning one, it opens a new field compared to previous works. So this work should be considered as a new baseline method in this newly proposed domain. So no matter how simple this method is, it should be considered as novel.

The only concern left is why people should care such a sub-field. However, I do not see this is a problem as we should encorage people to step out of the previous settings in my opinion. So I raise my score to accept.

---

> ### Author Response · Authors · 2022-11-19
> **Response to reviewer T2PL - Part 4-1 (4/4)**
>
> **W4. Please explain more clearly in section 5 about your training and searching setting when combining Meta, KD, few-shot learning and one-shot adaption, and NAS.**
>
> As we consider several important machine learning topics such as meta-learning, neural architecture search (NAS),  and knowledge distillation (KD) together to propose a novel framework that leverages the power of meta-learning to reduce the search cost for distillation-aware NAS problem, it could be quite confusing. Thus, to help the understanding of the reviewer, we explain the concept and limitations of existing approaches, the background of the follow-up study, the novelty of our method in comparison to existing methods, and detailed training and search settings below:
>
> **1. Meta-learning**
>
> While non-meta-learning methods (task-specific learning) have focused on learning a model that works well on a single task, meta-learning learns to generalize over task distribution, including multiple tasks, rather than a single task so that the meta-learned model can rapidly adapt to an unseen (new) task. However, most meta-learning methods have tackled **few-shot classification tasks** since applying meta-learning to a large-scale tasks is often impractical due to the heavy training cost of training over a large number of tasks.  On the other hand, meta-predictor-based NAS methods, including ours, apply meta-learning on more practical NAS tasks such as **(Da)NAS tasks**.
>
>
> **2. Predictor-based NAS methods and NAS task**
>
> * Before introducing meta-predictor-based NAS methods, we explain traditional predictor-based NAS methods, which are groundworks of meta-predictor-based NAS methods. Primitive (sampled-based) architecture search methods are very time-consuming as they sample an architecture candidate, train it with a few epochs, use initial validation performance as a metric for the candidate's final performance, and repeat the process for many architecture candidates to find the best architecture. To reduce the search cost, predictor-based NAS methods have proposed an architecture performance predictor that is simple multi-layer perceptrons, takes an architecture as input, and predicts its actual accuracy. By changing the time to train an architecture with few epochs into fast forwarding time of predictor, the predictor-based NAS methods can reduce the search time successfully.
> * However, to learn the predictor, they should collect thousands of (architecture-actual accuracy of the architecture) pairs by training thousands of architecture on a target task (pairs collecting time). Even if the total time (pairs collecting time + predictor training time + search time) is lower than the total time of sample-based primitive NAS methods, the total time is still excessively long. In other words, since predictor-based NAS methods only consider a **single** target task, for the new target task, we should collect thousands of (architecture-its actual performance on the target task) pairs again to search for optimal architecture on the new target task, which is highly impractical in real-world scenarios.

---

> > ### Author Response · Authors · 2022-11-19
> > **Response to reviewer T2PL - Part 4-4 (4/4)**
> >
> > **Meta-test setting (NAS on an unseen dataset and a teacher)**
> >
> > * At meta-test time, given a novel dataset and a teacher model trained on the dataset, we want to **rapidly search for a suitable student architecture to distill the knowledge from the teacher to the student. We utilize the meta-learned predictor to evaluate the given student architecture candidate by using the dataset and the teacher encoding. We randomly sample student architectures, select the architecture with the highest predicted performance using prediction models, and distill knowledge into the selected architecture with the teacher network and dataset.
> > * For a fair comparison between NAS methods, throughout the paper, we randomly sample 10,000 student architectures and select the top-1 student architecture found most optimal by each method. For detailed descriptions, please refer to **Section 4. Method**, **Figure 2**, **Generalization to Unseen Datasets of Section 5. Experiment**, and **Section 5.3** in the main paper.

---

> > ### Author Response · Authors · 2022-11-19
> > **Response to review T2PL - Part 4-3 (4/4)**
> >
> > **4. [Ours] Predictor-based Meta-NAS method for _DaNAS (KD, one-shot adaption)_**
> >
> > ---
> >
> > **Overall**
> >
> > * Even though the meta-predictor-based NAS methods can generalize well across multiple tasks, they search for an optimal architecture with good performance when performing scratch training on unseen target datasets. However, as explained above, an optimal architecture for scratch training and an optimal architecture for KD can be completely different. That means existing meta-predictor-based NAS methods are not distillation-aware and can be sub-optimal when considering DaNAS tasks. We design our meta-prediction model, suitable for DaNAS tasks, by considering the teacher and distillation-aware student embedding.
> > * Additionally, existing meta-predictor-based NAS methods for accuracy prediction propose to learn a cross-modal latent space of datasets and architectures (or networks) by performing amortized meta-learning, which is not a guided search strategy. They adopt this amortized meta-leaning strategy because it takes much time to get a few architecture-accuracy pairs which are few-shot samples needed for guided search on target tasks. In general, guided search, such as few-shot adaptation on the target task, requires the collection of few-shot architecture-accuracy pairs. To obtain even a single pair of architecture-accuracy, the architecture should be trained until it fully converges, which prevents rapid search at meta-test time.
> > * However, the DaNAS environment is different from the conventional NAS because we already have a trained model, the teacher network. We leverage this one teacher-network-accuracy pair (one-shot adaptation) which is always present in DaNAS environments, to make the meta-learned model adapt well to the target task. There is no need to wait to get the accuracy-architecture pair(s) for adaptation since we can get the accuracy of the teacher with a single forward pass.
> >
> > ---
> >
> >
> > **Meta-training setting**
> >
> > We divided meta-training settings in three steps:
> >
> > _1. Collecting meta-training database_
> >
> > To learn the proposed meta-predictor, we collect multiple DaNAS tasks. Each task consists of a dataset and a teacher trained on the dataset, a set of student architectures, and actual final accuracies of student architectures after distilling knowledge from the teacher network. We collected 8 tasks as the meta-training database and 2 tasks as the meta-validation database. For each task, a single dataset, a single teacher, 200 student architectures, and 200 actual distillation performances of the student architectures. For detailed descriptions, please refer to the **Section D. "Training details"** in the supplementary file in the revision.
> >
> > _2. Meta-learning our predictor_
> >
> > We sample a meta-batch containing multiple tasks for each iteration at the meta-learning phase. For each given task, the meta-predictor
> >     1) Takes a student architecture and teacher network,
> >     2) Encodes the architecture embedding of a student, functional embedding of the teacher, and functional embedding of the student whose parameters are remapped from the teacher, and
> >     3) Predict the distillation performance of the given student architecture.
> > By minimizing the predicted and actual accuracy of the student, we learn the proposed predictor. For detailed descriptions, please refer to **Section 4. Method**, **Figure 2** in the main paper, and **Section D. "Training details"** in the supplementary file.
> >
> > * One-shot adaptation
> >
> > Our meta-prediction model aims to generalize well over the DaNAS task distributions. In order to learn these task distributions well, for each given task at the meta-learning phase, we proceed with gradient-based guided adaptation with the information from the teacher network, which can be represented as the architecture embedding of the teacher and functional embedding based on the teacher. More specifically, the model rapidly becomes task-adaptive for each given task by minimizing the teacher's predicted accuracy and actual accuracy (performing inner gradient steps) with information from the teacher network. It is called _"one-shot adaption"_ because the model adapts to the given task with just _one_ teacher-accuracy pair, which always exists in DaNAS environments. For detailed descriptions, please refer to **"Adaptation Guided with Teacher-Accuracy Pair" of Section 4. Method** and **Figure 2** in the main paper.
> >
> >  _3. Adopting our model through meta-validation_
> >
> > We adopt a model with the lowest meta-validation loss on the meta-validation dataset as our meta-prediction model. After fixing our meta-prediction model, a meta-test phase is conducted. As in the meta-learning phase, the one-shot adaptation strategy with the teacher network-accuracy pair was used in the meta-test phase.

---

> > ### Author Response · Authors · 2022-11-19
> > **Response to reviewer T2PL - Part 4-2 (4/4)**
> >
> > **3. Meta-predictor-based NAS methods**
> >
> > * To overcome the generalization issue of existing predictor-based NAS methods on multiple tasks (datasets), meta-predictor-based NAS methods have proposed new dataset-conditioned architecture performance predictors, which support multiple datasets **with a single predictor** that can generalize to a new dataset. To this end, they learn dataset-conditioned predictors to generalize to multiple NAS tasks via meta-learning. Specifically, meta-predictor-based NAS methods have proposed novel (data)set encoders to use the dataset encoding as the predictors' input, allowing the predictor to make dataset-conditioned architecture performance predictions. They successfully generalized NAS frameworks to novel tasks, which involve unseen datasets and devices.
> > * On a new dataset, meta-predictor-based NAS methods can significantly reduce the search time by skipping the (architecture-accuracy) pairs collection part by simply encoding the architectures as well as the dataset, and forwarding them as inputs to the meta-learned predictor. From the meta-learning perspective, meta-predictor-based NAS methods are also novel as they have demonstrated the effectiveness of meta-learning on NAS tasks which is one of the practical problems compared with artificial few-shot classification tasks.
> >
> > * However, they are suboptimal if applied to a distillation-aware NAS task that aims to search for an optimal architecture when distilling the knowledge from the teacher.
> >
> > - Regarding the meta-learning algorithm of the meta-predictor for NAS:
> >
> > While existing meta-predictors for NAS have proposed the new predictor structure enabling dataset-conditioned predictions and have learned the single predictor over task distribution, none of them use gradient-based guided adaptation algorithm with few-shot samples, which can enhance the adaptation performance of the predictor on a new dataset.

---

> ### Author Response · Authors · 2022-11-19
> **Response to reviewer T2PL - Part 3 (3/4)**
>
> **W3-1. The position of references mentioning Tables and Figures in the paper are several pages away from the Tables and Figures.**
>
> * Thank you for the comment. We modified the paper's layout so that the references mentioning Tables and Figures are located on the same page or the next page of related Tables and Figures.
>
> **W3-2. What is the unit of the results in the main results are in Table 2?**
>
> * As described in the **Generalization to Unseen Dataset of Section 5. Experiment** in the previous version and **Section 5.1.** in the revision, the results in Table 2 are Spearman's rank correlation coefficients between **rankings of the architectures by their actual performances with knowledge distillation** and **rankings by their predicted performance with knowledge distillation**. Spearman's rank correlation coefficient is a non-parametric measure of rank correlation that does not have a unit and represents how similar the rankings by each variable are. For improved clarity, we included this description in the caption of Table 2 as well as **Section 5.1** in the revision.
>
> **W3-3. What are the model components: Set. PR. and Gui.?**
>
> * These refer to the components of the predictor input. In more detail, “Arch.” denotes taking an architecture embedding as an input, “Set.” denotes using a dataset embedding as an input, “PR.” denotes taking the functional embedding of the student architecture candidates with remapped parameters from the teacher, and “Gui.” denotes the use of gradient-guided meta-learning strategy. To reduce the confusion with these abbreviations (Arch., Set., PR., Gui.,), we included the aforementioned descriptions in **Section 5.1** and modified them a bit as follows: Set. $\rightarrow$ Set Enc. / PR. $\rightarrow$ Pr. Remap. / Gui. $\rightarrow$ Guide
>
>
> **W3-4. Please explain more in the caption (Tab 2,3,4,5).**
>
> * Following your comment, we added detailed captions for Tables 2, 3, 4, and 5 in the previous version. Each Table corresponds to Tables 2, 7, 3, and 4 in the revision. Please refer to the revision.
>
> **W3-5. What is collecting time in figure 4?**
>
> * The “collecting time for Ours (student)” in Figure 4 of the previous version (Figure 3 in the revision) refers to the time required to obtain several (architecture-performance) pairs in the guided meta-learning strategy. The “Number of samples” means the number of these pairs. In order to obtain one pair, knowledge distillation for each network must be performed until full convergence, since the performance must be obtained on a fully converged model. In a Distillation-aware NAS task, there always exists a teacher network that has been fully trained, and thus if we use the guided meta-learning strategy with the (architecture-performance) pair of the teacher network as we suggested, we need only a single forward pass to obtain the teacher’s performance. The collecting time is the time taken by this forward pass to obtain the accuracy of the teacher network (“collecting time for Ours (teacher)”  in Figure 4). We included descriptions of this terminology in the caption of Figure 4 (Figure 3 in the revision).
>
> ***
> To enhance the clarity of the paper, we incorporated the suggestions in the revision. We thank the reviewer for the valuable comments.

---

> ### Author Response · Authors · 2022-11-19
> **Response to reviewer T2PL - Part 2 (2/4)**
>
> **W2-1. typo: many sentences are too difficult to understand, eg on page 2: we propose a distillation-aware task encoding function that remaps parameters from the teacher to the student and considers the output from the embedding network of the remapped student as the embedding to estimate the teacher network’s impact on the final accuracy of the distilled model.**
> * Following your comments, we revised the sentences to introduce the distillation-aware task encoding function as follows: “First, we propose a distillation-aware task encoding function that considers the output from the student whose parameters are remapped from the teacher to estimate the teacher network's impact on the final performance of the distilled student network.”. Additionally, we clarified the details of the distillation-aware task encoding function in **Section B. Implementation Details** of the supplementary material . Please check the updated texts in the revised version of the paper.
>
> **W2-2. The writing is really hard to follow, especially when combing a lot of new concepts. More, concepts need to be expiated clearly for example the setting of the few-shot learning under this paper's scenario.**
> * We sincerely appreciate your suggestions. We updated the paper by rewriting the parts that were difficult to follow and new concepts. Also, we explained our method and experiment settings in more detail in Response Part 4 (4/4). Please check the updated paper.
>
> ***
> We believe that we have largely enhanced the clarity of the paper thanks to your suggestion. Please check them and see if there is any remaining concern on the clarity or the quality of writing.

---

> ### Author Response · Authors · 2022-11-19
> **Response to reviewer T2PL - Part 1 (1/4)**
>
> Thank you for your constructive comments. We address your comments below:
>
> **W1-1. Some logic is hard to follow, for example, in Intro, conventional NAS is sub-optimal for searching students under KD because the optimal student distilled from the teacher is different from an architecture trained from scratch.**
> * As a proof of concept, we conducted the following experiment on the subset of TinyImageNet. For the two architectures, we compared the performances of scratch training and knowledge distillation with a pre-trained teacher network. As shown in the Table, the performance of scratch training with only ground-truth labels and the performance of knowledge distillation can be reversed. That means an architecture optimized for knowledge distillation differs from that optimized for ground-truth labels, and good performance on scratch training does not guarantee good performance of knowledge distillation.
> * However, as you mentioned in your review, conventional NAS do not consider knowledge distillation components such as the teacher network trained on the target dataset. They search for an optimal architecture that performs well when performing scratch training. Therefore, the optimal architecture from conventional NAS is sub-optimal for tasks that consider knowledge distillation. That is what the following sentence meant: "conventional NAS is sub-optimal for searching students under KD because the optimal student distilled from the teacher is different from an architecture trained from scratch." in the **Section 1. Introduction**. Reflecting on your review, we have explained in more detail why conventional NAS is sub-optimal in the knowledge distillation scenario in the revision. We could improve our paper based on your feedback.
>
> | Training Pipeline      |  Teacher | Arch. 1 Acc. (%) | Arch. 2 Acc. (%) |            Comparison           |
> |------------------------|:--------:|:---------------------------:|:---------------------------:|:-------------------------------:|
> | Scratch Training       |     -    |            42.97            |            46.84            | Arch. 1 < Arch. 2 |
> | Knowledge Distillation | ResNet34 |            49.53            |            47.63            | Arch. 1 > Arch. 2 |
>
> **W1-2. In the abstract, the author claims that meta-learning NAS is sub-optimal for DaNAS. But from my perspective, they are designed originally for solving the generalization issue for NAS, and of course, they are not the optimal choice for DaNAS. The author claims they are not optimal just because they want to adopt meta-learning in DaNAS and find that they do not consider teacher aspect so they are not optimal.**
> * As shown in the above Table in **W1-1**, the suitable architectures for scratch training and KD training are different. However, while the teacher model largely affects the actual performance of the architecture trained with knowledge distillation, existing meta-accuracy performance predictors do not consider the teacher model at all. In other words, as you mentioned, the previous meta-learning-based accuracy predictors designed for the NAS problem are unsuitable for DaNAS task. Thus we needed to design a novel meta-predictor for DaNAS task. Moreover, how to design the predictor to consider the distillation process is an important and non-trivial problem for DaNAS. We tackle this problem by proposing _a novel distillation-aware task encoding_. Such distillation-aware task encoding is based on the parameter remapping from a specific teacher to a student and the functional embedding of the remapped student along with the functional embedding of the teacher.
> In addition, we propose a _gradient-based one-shot adaptation scheme with the teacher network and its accuracy pair_, and we are the **first** to validate the effectiveness of the gradient-based adaptation on Meta-NAS problem for accuracy prediction.
> The idea of the pre-trained teacher network-its accuracy as an adaptation ingredient is simple yet effective for DaNAS tasks as it enables rapid architecture search by removing the large training time to obtain architecture-its accuracy pairs for adaptation (collecting time) on a given task. While the existing meta-accuracy performance predictors have not leveraged the power of gradient-based adaptation on NAS task due to the large collecting time.
> In sum, ours is the first work to validate the effectiveness of gradient-based adaptation of meta-accuracy-predictors on NAS tasks and the first work to tackle the generalization issue on DaNAS tasks, including the large-scale dataset and datasets from the out-of-distribution domain (please refer to the revision). In contrast, most DaNAS works have validated their frameworks on a very limited number of datasets, such as ImageNet-1K, Cifar-10, and Cifar-100. That is, ours is the first lightweight and rapid search method on target datasets while most DaNAS methods are extremely costly to be used for multiple datasets.

---

> ### Comment · Reviewer_T2PL · 2022-11-26
> **Update**
>
> As authors submitted a super-long response, I need some time to carefully read them and revise my rating. I plan to do it on Nov 27.
>
> Best
> Reviewer T2PL

---

> ### Author Response · Authors · 2022-11-27
> **We appreciate the Reviewer T2PL**
>
> We appreciate the Reviewer T2PL for your feedback and for pointing out the novelty of our work which opens a new field compared to the previous distillation-aware NAS and meta-learning. Following the comments suggested by Reviewer T2PL, we improved the clarifications of our motivation, contributions, and details of experiments in the revision. These things strengthen our paper, and we thank the Reviewer T2PL again for the active discussions and thoughtful comments.

---

### Author Response · Authors · 2022-11-19
**Summary of the responses from all reviewers**

We sincerely appreciate your time and effort in reviewing our paper, as well as the positive feedback that the motivation is interesting (T2PL), the problem is new (L5tN), the approach is novel (T2PL, L5tN), and well-motivated (FiPx), the results are promising (L5tN), and validation suggests the proposed method is effective (FiPx). We have responded to the individual comments from the reviews below, and believe that we have successfully responded to most of them. We have included the discussion and results of the suggested experiments in the revision. Here we briefly summarize the updates we have made to the revision:
* **FiPx**: Adding experiments on datasets in **different domains** including fine-grained datasets (CUB, SFC), out-of-distribution datasets (DTD - texture images, EuroSAT - satellite images, ISIC - gray medical images about skin lesions, ChestX - X-ray images) to demonstrate the practical values of the proposed method.
***
* **FiPx**: Adding experiments on datasets consisting of **large images** with 224 $\times$ 224 pixels, and 256 $\times$ 256 pixels to demonstrate the practical values of the proposed method.
***
* **FiPx** and **L5tN**: Adding experiments on **large-scale datasets** such as ImageNet-1K to demonstrate the practical values of the proposed method.
***
* **FiPx**: Adding experiments with **larger** teachers and students with more parameters to demonstrate the practical values of the proposed method.
***
* **FiPx**: Adding discussions of the architecture configurations and efficiencies (MACs) obtained by the search process.
***
* **FiPx** and **L5tN**: Improving clarity by updating the description of the parameter remapping.
***
* **L5tN**: Improving clarity by updating the description of the encoding part.
***
* **T2PL**: Improving clarity by explaining in more detail why conventional NAS and meta-NAS methods are sub-optimal in the knowledge distillation scenario.
***
* **T2PL**: Improving clarity by rewriting the parts that were difficult to follow and contained a lot of new concepts.
***
* **T2PL**: Improving clarity by modifying the paper's layout so that the references mentioning tables and figures are located on the same page or the next page of related tables and figures.
***
* **T2PL**: Improving clarity by adding detailed descriptions about abbreviations and metrics used in the experiments.
***
* **T2PL**: Improving clarity by adding detailed captions for tables and figures.
***
We appreciate your constructive suggestions, as we found the new discussions and experimental results to be highly valuable, largely strengthening our paper. Finally, we emphasize again that 1) exploring a challenge of generalization on unseen DaNAS tasks and 2) proposing a meta-learning framework to tackle this issue are both highly novel contributions with potentially significant practical impacts.

---

### Comment · Area_Chair_AR8S · 2022-12-06
**Updated PDF?**

Dear authors,

Thank you for your comprehensive responses.
Despite the fact that you report many additional results in your response, you unfortunately missed the opportunity to update your paper during the first two weeks of the rebuttal. It is thus very hard to judge to which degree these additional results would improve the paper.
I would nevertheless like to give you the chance to do so and therefore encourage you to provide a link to an anonymous PDF (e.g., in an anonymous code repository) that would reflect what the paper would look like with your changes.

I would also like to remark on the extraordinary amount of reminders you have been sending. All reviewers have been responsive, and there has been much more interaction than for most other papers. But the reviewers also juggle several different papers and do this purely as voluntary service, so at some point the n-th reminder is really too much. I would thus encourage you to instead focus your efforts on the updated PDF above. Thanks!

Best,
AC

---

> ### Author Response · Authors · 2022-12-06
> **Our uploaded revision includes all results in the responses.**
>
> Dear Area Chair,
>
> Thank you for your time and effort in meta-reviewing our paper.
> We would like to politely let you know that we already updated the paper and uploaded the revision in the first two weeks. Our revised version of the paper, that is uploaded to the openreview system faithfully **includes all additional results** that we provided in our responses.
>
> You can check the differences between our initial submission and the revision, highlighted with colors via openreview system as follows:
>
> *(Show Revisions - Compare Revisions (button on the top right) - Check the last version and the version named 'Meta-prediction Model for Distillation-Aware NAS on Unseen Datasets' - View differences (button on the top right))*
>
> Please only consider the documents with the PDF icons.

---

### Decision · Program_Chairs · 2023-01-20

**Decision:**

Accept: notable-top-25%

**Justification For Why Not Higher Score:**

Too many of the results were produced in rapid cycles during the rebuttal period; there is thus a certain risk of them not having been as carefully checked as results that were in the original paper and had some time to settle.
Also, the authors only got so much interaction (and thus an average increase of scores of more than 2 points) because they sent 10+ reminders, which implicitly pressured the reviewers to iterate several times; while this iteration is a valuable possibility of the OpenReview rebuttal process, it is not sustainable for the community that everyone requests it, and I would not reward it with an oral.

**Justification For Why Not Lower Score:**

I would also be fine if the PCs decide to downgrade to a poster if there are good reasons for that looking at the big picture.
However, please don't go by the scores of 8,8,3 in the system but rather the effective scores 8,8,6 (the rejecting reviewer didn't update their score but said the paper is above the acceptance threshold for them now). So, the effective average score is 7,33, which I would expect to be in the spotlight zone.

**Metareview: Summary, Strengths And Weaknesses:**

This paper proposes to make NAS distillation-aware by searching for a student architecture that performs best when distilling from a given teacher, with a meta-learning component that allows generalization to new datasets.
During the discussion, the authors were able to produce several new results to satisfy various concerns by the reviewers, and as a result two reviewers increased their score to 8. The third reviewer, FiPx, quoted the ICLR FAQ that new experiments should not significantly change the content, but did not take into account that these experiments were done specifically to answer the questions of reviewers (which is perfectly OK). The reviewer stated (in private message) that they would rate the submission as above the threshold if the revision is allowed. This is the case (at least in my interpretation of the guidelines, but I believe this aligns with the community's interpretation), and therefore, the effective scores are 8,8,6.
Furthermore, meta-learning is a very understudied area of NAS, making the work very timely. I therefore would recommed the paper for a spotlight.

**Note From Pc:**

if the above contains the word "oral" or "spotlight" please see: "oral" presentation means -> notable-top-5% and "spotlight" means -> notable-top-25%. As stated in our emails, we are disassociating presentation type from AC recommendations